# Ultra-fast and accurate electron ionization mass spectrum matching for compound identification with million-scale *in-silico* library

Qiong Yang[1,3], Hongchao Ji[2,3], Zhenbo Xu[1], Yiming Li[1], Pingshan Wang [1], Jinyu Sun[1], Xiaqiong Fan[1], Hailiang Zhang[1], Hongmei Lu [1] ✉ & Zhimin Zhang [1] ✉

Spectrum matching is the most common method for compound identification in mass spectrometry (MS). However, some challenges limit its efficiency, including the coverage of spectral libraries, the accuracy, and the speed of matching. In this study, a million-scale *in-silico* EI-MS library is established. Furthermore, an ultra-fast and accurate spectrum matching (FastEI) method is proposed to substantially improve accuracy using Word2vec spectral embedding and boost the speed using the hierarchical navigable small-world graph (HNSW). It achieves 80.4% recall@10 accuracy (88.3% with 5 Da mass filter) with a speedup of two orders of magnitude compared with the weighted cosine similarity method (WCS). When FastEI is applied to identify the molecules beyond NIST 2017 library, it achieves 50% recall@1 accuracy. FastEI is packaged as a standalone and user-friendly software for common users with limited computational backgrounds. Overall, FastEI combined with a million-scale *in-silico* library facilitates compound identification as an accurate and ultra-fast tool.

Mass spectrometry (MS) is a convenient, highly sensitive, and reliable method for the analysis of complex mixtures, which is vital for life sciences fields such as metabolomics and proteomics, and organic synthesis in chemistry[1]. High throughput identification of compounds in these complex samples is enabled by MS generation of thousands to millions of spectra, from which many thousands of compounds are typically identified by matching the experimental MS spectra against libraries containing a list of known molecular masses and fragmentation patterns. However, the vast majority of compounds in an MS experiment cannot be identified due to limited coverage in existing libraries. For example, only approximately 20% of compounds based on gas chromatography-mass spectrometry (GC-MS) can be identified

by spectrum matching with current libraries[2]. If the sample contains compounds that are absent from spectral libraries, the correct identification of them is nearly impossible because of false negatives. For false positives, some additional information can be used for effective filtering, such as molecular weight, retention index, and domain knowledge. By contrast, there are seldom remedies for a false negative. Therefore, it can be said that the cost of making a false negative is much higher than a false positive when identifying compounds. The coverage of the libraries needs to be increased as much as possible to avoid false negatives.

Many efforts have been made to solve this problem. Common methods include acquiring experimental spectra of standards and

[1]College of Chemistry and Chemical Engineering, Central South University, Changsha 410083 PR, China. [2]Agricultural Genomics Institute at Shenzhen, Chinese Academy of Agricultural Sciences, Shenzhen 518120 PR, China. [3]These authors contributed equally: Qiong Yang, Hongchao Ji.
✉ e-mail: hongmeilu@csu.edu.cn; zmzhang@csu.edu.cn

generating *in-silico* spectra. The quantum chemistry electron ionization MS (QCEIMS) is a typical method that combines Born-Oppenheimer molecular dynamics (BOMD) with statistical sampling to predict mass spectra[3]. However, the experiment and quantum chemistry strategy are costly and time-consuming to generate spectra for thousands of molecules. As a matter of fact, there are >111 million molecules with structures in PubChem[4] and two million bioactive molecules in ChEMBL[5]. There are only 0.27 million compounds with electron ionization mass spectra in NIST 2017 library[6]. Recently, the molecule structure information combined with machine learning has become a burgeoning alternative for predicting *in-silico* spectra. For example, competitive fragmentation modeling for electron ionization (CFM-EI) applies a probabilistic generative model to predict the electron ionization mass spectrum (EI-MS) from SMILES[7]. The neural electron-ionization mass spectrometry (NEIMS) method takes the extended circular fingerprints (ECFPs) of molecules as inputs and applies fully connected neural networks to predict the spectra[8]. It can quickly and easily generate large-scale *in-silico* spectra from molecular structures, thus extending the chemical space and immensely increasing the coverage compared to experimental and quantum chemistry methods.

With a large-scale *in-silico* library, another challenge is how to rapidly match query spectra with millions or even tens of millions of spectra while ensuring the accuracy of compound identification. The accuracy of spectrum matching partly depends on whether the metric correctly reflects the similarity between the query spectrum and reference spectrum[9,10]. The weighted cosine similarity (WCS) is the most common method in mass spectrometry[10]. Other metrics are also employed, including Euclidian or Hamming distance, probability-based matching[9], weighted average ratio[11], and neutral-loss matching[12]. Matyushin et al. utilize a convolutional neural network (CNN) to search the EI-MS library and achieve better accuracy than the default method of NIST MS Search software[13]. The above methods can be regarded as the exact nearest neighbor search (NNS) methods. They are particularly time-consuming for matching large-scale libraries because of the scalability issue[14]. When searching a large-scale library, the approximate nearest neighbor search (ANNS) methods are preferred,

including hierarchical navigable small world graphs[14], inverted file[15], locality-sensitive hashing[16], and product quantization[17]. These methods achieve significantly improved speedup by allowing a small number of errors in the searching procedure.

Here, we develop a strategy to expand the coverage of spectral libraries and propose an efficient method to search large-scale libraries. The overview of the proposed method is illustrated in Fig. 1. The 2,146,690 molecular structures come from the *f*-NIST and *f*-ChEMBL, which are obtained by filtering and deduplicating the NIST 2017 and ChEMBL 28[5,18], respectively (Supplementary Fig. 1a). The predicted EI-MS spectra of these 2,146,690 molecules included in the *in-silico* library (Supplementary Fig. 1b) are generated by NEIMS using their molecular ECFPs as inputs. An ultra-fast and accurate spectrum matching (FastEI) was proposed to match the million-scale *in-silico* library efficiently. It consists of Word2vec spectral embedding for better accuracy and the hierarchical navigable small-world graph (HNSW) for faster spectrum matching[14]. We have demonstrated the performance of FastEI in terms of identification accuracy and spectrum matching speed on a test set and an extra test set. Over ten thousand measured spectra of molecules in the test set were collected from NIST 2017. The extra test set includes ten synthetic organic compounds with measured spectra, which are not included in the NIST 2017. In general, FastEI is promising for compound identification along with the large-scale *in-silico* library.

## Results

### Million-scale *in-silico* EI-MS library

To solve the *coverage problem* in compound identification, a million-scale *in-silico* library was generated using NEIMS to get predicted EI-MS spectra from molecular ECFPs. The NEIMS model was downloaded from its official repository without any further operations.

Firstly, to evaluate the predictive performance of NEIMS, the weighted cosine similarity (WCS) distribution between the predicted and experimental spectra in the test set is shown in Supplementary Fig. 2. The similarities between predicted and experimental spectra of 78% of molecules are greater than 0.7. It demonstrates that NEIMS can accurately predict the EI-MS spectra for molecules. Meanwhile, Fig. 2a

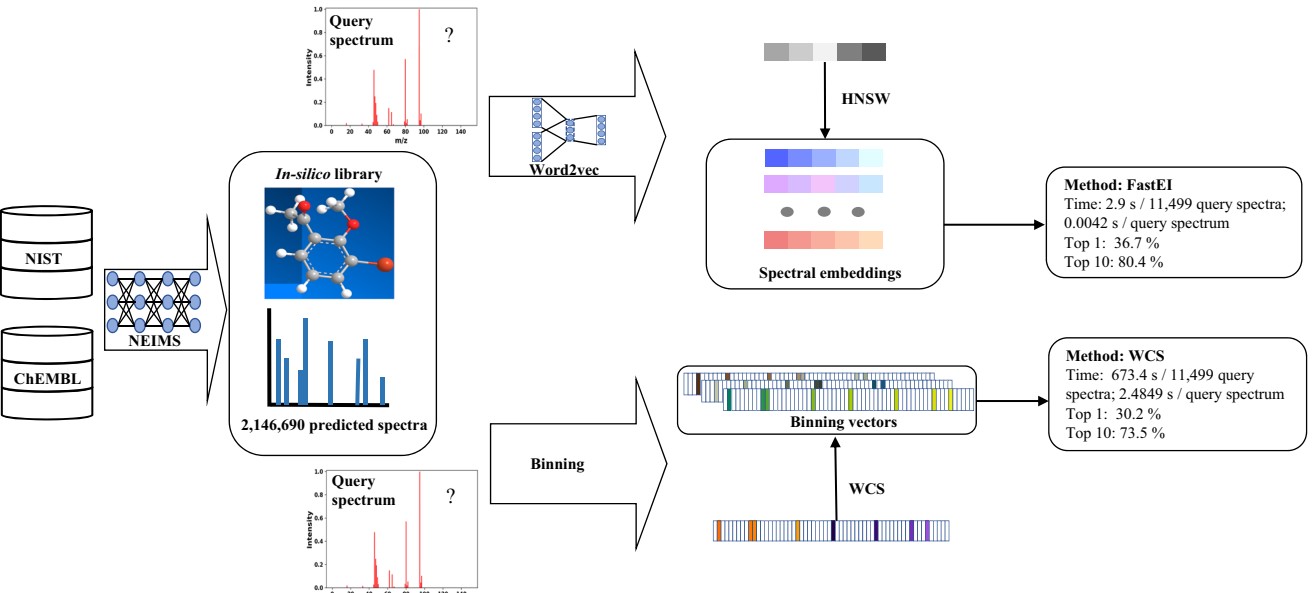

**Fig. 1 | Flowchart of FastEI and WCS methods.** First, a large-scale *in-silico* library is generated from the NIST and ChEMBL datasets. For the querying of spectra, FastEI (top branch) uses Word2vec to transform the spectrum into spectral embeddings. These embeddings are given a Hierarchical Navigable Small-world Graph (HNSW)

index, which is used for retrieving similar candidates from the library. Comparatively, classical binning methods (bottom branch) divide the spectra in bins and compare them with the ones in the library, most commonly using the weighted cosine similarity (WCS) as the measure.

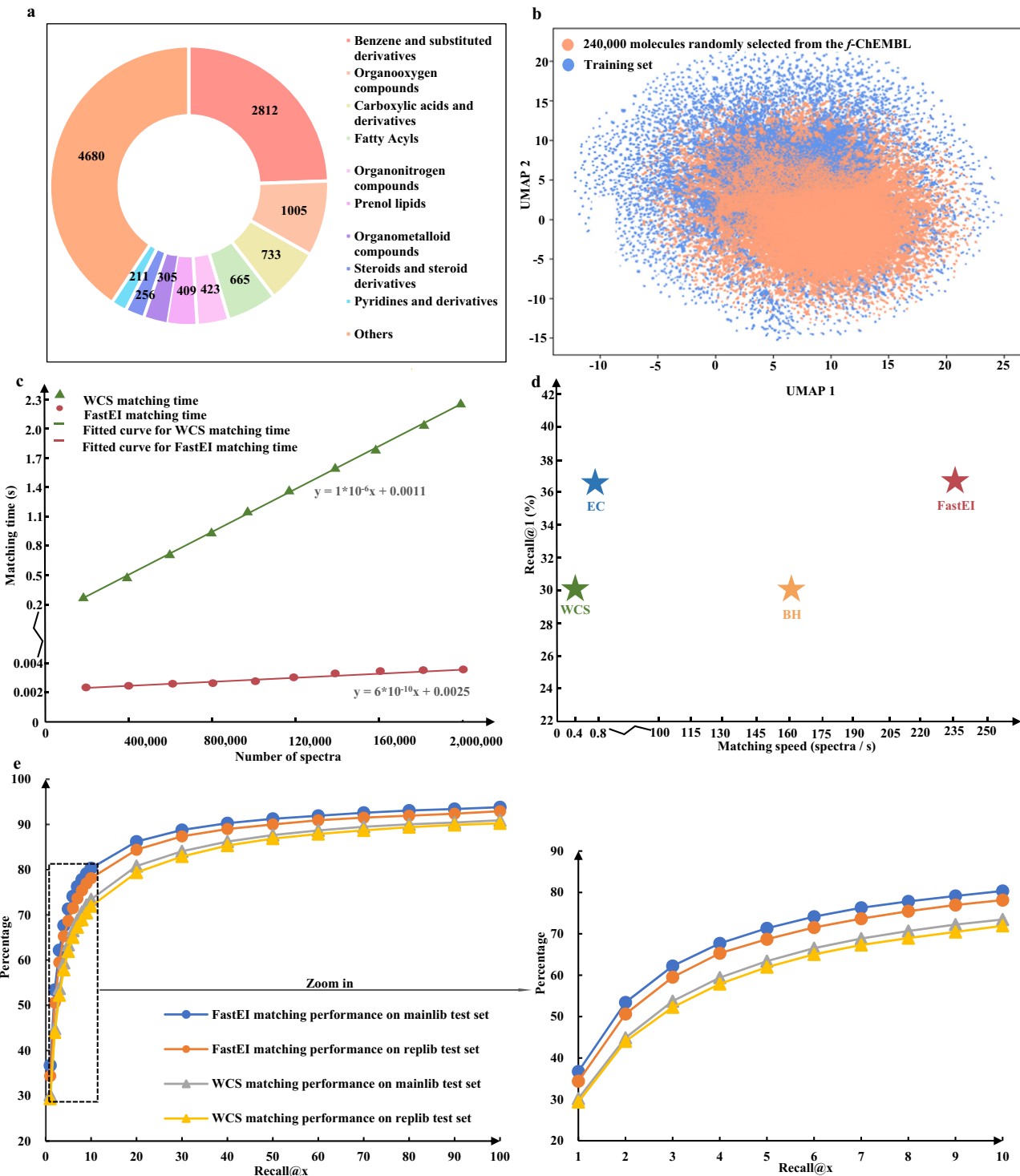

**Fig. 2 | Information of datasets and performance of FastEI. a** The molecule classes predicted by ClassyFire for the test set. **b** The visualization of the ECFPs of 240,000 molecules randomly selected from *f*-CHEMBL and 232,826 molecules from the training set by UMAP. **c** The spectrum matching time of FastEI and WCS on libraries with different sizes. **d** The contribution of Word2vec embeddings and

HNSW to FastEI (WCS weighted binning + cosine similarity, EC embeddings + cosine similarity, BH weighted binning + HNSW, FastEI embeddings + HNSW). **e** The performance of FastEI and WCS on the test set in terms of recall rates at different top *x* levels. Source data are provided as a Source Data file.

shows the classes of molecules in the test set predicted by ClassyFire[19]. It demonstrates that the test set covers vast chemical structures. On the other hand, to validate the applicability of the NEIMS model for predicting mass spectra of molecules in *f*-ChEMBL, the uniform manifold approximation and projection (UMAP)[20,21] was used to visualize the extended circular fingerprints (ECFPs) of molecules from

the training set and *f*-ChEMBL. A total of 240,000 molecules (about 12%) were randomly sampled from the *f*-ChEMBL dataset to reduce the computational cost without losing representativeness. Then, RDKit[22] was used to calculate the ECFPs of these molecules. The high-dimensional ECFPs (1024 bits) were transformed into two-dimensional representations by UMAP for visualization. As shown in

**Table 1 | The spectrum matching performance on the *in-silico* library**

| Method | Recall@1 (%) | Recall@10 (%) | Matching speed (s per query spectrum) |
|---|---|---|---|
| FastEI | 36.7 | 80.4 | 0.0042 |
| WCS | 30.2 | 73.5 | 2.4849 |
| FastEI + mass filter[a] | 45.3 | 88.3 | – |
| WCS + mass filter[a] | 37.1 | 81.6 | – |

[a]The mass filter was set to 5 Da of the query molecule's mass.

**Table 2 | The matching performance on the expanded library**

| Method | Recall@1 (%) | Recall@10 (%) | Matching speed (s per query spectrum) |
|---|---|---|---|
| FastEI | 36.4 | 79.9 | 0.0046 |
| WCS | 29.7 | 72.9 | 2.6368 |

Fig. 2b, the training set completely covers the molecules randomly sampled from *f*-ChEMBL in the chemical space. Therefore, it is reasonable to apply the NEIMS model to predict the mass spectra of molecules in the *f*-ChEMBL. Finally, 2,146,690 predicted spectra were generated to build the *in-silico* library, adopting ECFPs of molecules from *f*-NIST and *f*-ChEMBL as inputs of the NEIMS model (Supplementary Fig. 1b).

### Spectrum matching performance of FastEI

The accuracy and speed of FastEI were compared with WCS (Supplementary Note 1) on the test set. The measured spectra of the test set were collected from the NIST 2017 main library. The comparison results of FastEI and WCS are shown in Table 1. We can find that the run time of FastEI is 0.0042 s per query spectrum, and that of WCS is 2.4849 s per query spectrum. FastEI is about 592 times faster than WCS when matching one spectrum.

In analyzing complex systems, such as untargeted metabolomics[23] and environmental science, it is necessary to quickly identify unknown compounds in bulk based on mass spectrometry. For such cases, the speed of spectrum matching methods becomes more critical, especially if the *in-silico* library is large too. A total of 11,499 experimental spectra in the test set were matched with the *in-silico* library at once by FastEI and WCS methods. As shown in Fig. 1, only 2.9 s were needed to get the matching results by FastEI. In contrast, about 11.2 min were required by WCS. FastEI is 232 times faster than WCS in large-scale spectrum matching. For the matching accuracy (Table 1), FastEI achieves a matching result with 36.7% recall@1 and 80.4% recall@10. After applying a 5 Da mass filter to the spectrum matching result, FastEI achieves 45.3% recall@1 and 88.3% recall@10. By contrast, the matching accuracy of FastEI is better than WCS with 6.5 and 6.9 percentage points at recall@1 and recall@10, respectively, when without the mass filter. With the 5 Da mass filter, the difference in the matching accuracy of the two methods (FastEI and WCS) is 8.2 and 6.7 percentage points in recall@1 and recall@10, respectively. In short, the FastEI method outperforms the WCS method in matching speed and accuracy. In practice, this mass filter could be set according to the practical requirement[24].

In order to verify the impact of the library size on the spectrum matching speed, one mass spectrum was randomly selected from the test set to calculate the matching time of FastEI and WCS methods on the libraries with different sizes. The results are shown in Fig. 2c. It can be seen that the matching time of WCS linearly increases with the number of spectra in the library, and the time of FastEI hardly increases with the library size. As the number of spectra in the *in-silico* library

increases, this advantage becomes more and more apparent. When the number goes up to ~2,000,000, the matching speed of FastEI is five hundred times faster than WCS. This unique characteristic should be attributed to the hierarchical graph structure of HNSW for efficient graph traversal.

### Contribution of Word2vec embeddings and HNSW

As shown in Fig. 2d, a comprehensive study is conducted to demonstrate the contribution of Word2vec embeddings and HNSW in the FastEI method. Speed in Fig. 2d is defined as how many mass spectra per second can be identified. Accuracy is the recall@1 of the identified spectra in the test set. FastEI can match 36.7% spectra with recall@1 at 238 spectra per second speed. It can be seen from Fig. 2d that FastEI is the best in terms of both speed and accuracy. Meanwhile, the accuracy of the EC (embeddings + cosine similarity) method, which is formed from the binning in WCS replaced by embeddings, is improved to a level comparable to that of FastEI, but with a similar matching speed to WCS. The matching speed of BH (weighted binning + HNSW) method, which is formed from the cosine similarity in WCS replaced by HNSW, is improved partly with the same accuracy as WCS. Results show that the Word2vec embeddings mainly improve accuracy. Meanwhile, the HNSW boosts the matching speed. The HNSW index is an efficient data structure (the hierarchically multi-layer graph) to store and organize data. It can significantly improve the speed of ANNS. In brief, the FastEI method achieves high accuracy and speed by combining Word2vec embeddings and HNSW.

### Applicability of FastEI to spectra from different sources

To evaluate the generalizability to different mass spectral sources and qualities, the spectra of the test set were collected from two different sources, the NIST 2017 main library (*mainlib*) and replicate library (*replib*). The spectra of the *mainlib* test set are from *mainlib*, a collection of the "best spectrum" for each compound based on human evaluation. In contrast, the spectra in the *replib* test set are from the NIST 2017 replicates library, which is a collection of noisier spectra due to inconsistency of sources, such as instrument type, experiment condition, and laboratory. The spectrum matching performance of the FastEI and WCS methods was evaluated by the *mainlib* test set and the *replib* test set. As shown in Fig. 2e, the spectral matching performance of FastEI significantly outperforms WCS by an average of 6 percentage points in recall rates at different top x levels. The matching results of the *replib* spectra are basically the same as those of the *mainlib* spectra. It indicates that FastEI is a robust method.

### Incremental expansion of the library

In practical application, mass spectral libraries often need to be expanded due to the introduction of new compounds. It is easy to implement by predicting the mass spectra of these new compounds using the existing NEIMS model and adding them to the *in-silico* library. The predicted spectra are converted into spectral embeddings by the trained Word2vec model. Then, these spectral embeddings are added to the original HNSW index to build the expanded HNSW index. This simple operation can complete the update of FastEI with the expansion of the mass spectral library.

A total of 106,526 predicted spectra of new molecules from *f*-HMDB and the extra test set were added into the *in-silico* library to get the expanded library (Supplementary Fig. 1b). The expanded library includes 2,253,216 molecules and their predicted spectra. The experimental spectra (collected from the NIST 2017 main library) of molecules in the test set were used again to evaluate the identification performance of FastEI and WCS on the expanded library. As shown in Table 2, the recall@1 and recall@10 of FastEI are 36.4% and 79.9%, respectively. The recall@1 and recall@10 of WCS are 29.5% and 72.9%, respectively. It can be seen from Tables 1 and 2 that expanding the *in-silico* library with nearly 100,000 molecules reduces the recall rate of

0.3% of recall@1 and 0.5% of recall@10 in the FastEI method, respectively. It reduces the recall rate of 0.5% of recall@1 and 0.6% of recall@10 in the WCS method, respectively. The average run time (s) for matching one molecule against the expanded library of FastEI and WCS is 0.0046 s and 2.6368 s, respectively. The results in Table 2 indicate that the expansion of the *in-silico* library with nearly 100,000 molecules slightly reduces the accuracy of the matching performance and slightly increases the matching time. The slight decrease in the matching accuracy should be attributed to the expanded library having more compound structures than the *in-silico* library. As shown in Supplementary Figs. 3, 7 new classes (ClassyFire) of compounds appear in the expanded library compared to the *in-silico* library. For a given query spectrum, there are more compounds with similar structures or similar spectra to the target compound in the expanded library. In short, the expansion of the *in-silico* library is simple and has little impact on the performance of FastEI.

### Unknown compound identification

GC-MS is often used to monitor the reaction progress[25,26] and analyze reaction samples[27] to identify reactants, reaction intermediates, and products in organic synthesis. However, it is difficult to identify compounds absent from the NIST 2017 library through spectrum matching due to the limited compound coverage. To demonstrate the advantage of the million-scale *in-silico* library and the accuracy of FastEI, 10 compounds beyond the NIST 2017 library were collected from an organic laboratory. Their structures are shown in Fig. 3a. Compounds 1, 2, 4, and 5 are common aromatic compounds with different substituents on the benzene rings. If these 10 molecules are identified by matching the NIST 2017 library directly, the correct result cannot be achieved. Because there are no correct molecules in the NIST 2017 library, and all the candidates are false positives. With the aid of the large-scale *in-silico* library, these compounds can be mostly identified by FastEI. Their ranks are shown in Fig. 3a. It can be seen that their top 1 accuracy is 50%, and the top 10 accuracy can reach 70%. The compound with a rank >100 is only compound 10, which has a 12-carbon long-chain substituted on the benzene ring. Its MS spectrum is shown in Fig. 3b. It can be seen that most peaks are concentrated in m/z < 200. Many chain alkanes candidates are ranked before the target compound, possibly due to the existence of a long-chain structure (Fig. 3c). Overall, these results show that FastEI with the large-scale *in-silico* library can achieve good identification results for molecules beyond the NIST 2017 library. Expanding the spectra library by machine learning-based methods is beneficial to compound identification through reducing the proportion of false negatives.

### FastEI software

We provide an integrated spectrum matching software with the *in-silico* library and matching algorithms. The core functions of FastEI are implemented with Python programming languages, and the graphical user interface (GUI) is based on the Qt framework. The software can run on Windows 7, 10, and 11 operating systems. As shown in Fig. 4, the user interface is simple and friendly. For the universality, when the software is opened, the expanded library is loaded by default (Fig. 4a). Meanwhile, it can switch between different libraries without restarting the software. Researchers can also build tailored libraries according to their research aims.

Users can click the Query button to select experimental spectra for identification. As shown in Fig. 4b, the query spectra are loaded. The library matching results are available immediately after the query spectra are loaded. Users can click on any query spectrum to see its candidates list in the upper part of Fig. 4c. Then, users can click any compound in the candidate list to show its spectrum in the *in-silico* library against the measured query spectrum in the lower part of Fig.4c along with its molecular structure. Figure 4d is a bar that displays the

progress of data loading and spectrum matching. The installation package of FastEI is available at https://github.com/Qiong-Yang/FastEI/releases. Overall, FastEI is a convenient and easy-to-use software for searching a million-scale *in-silico* library quickly and accurately.

## Discussion

Spectrum matching is one of the vital steps for identifying compounds in metabolomics, organic synthesis, biology, and others. In this study, an ultra-fast and accurate spectrum matching method, FastEI, was proposed. Firstly, a million-scale *in-silico* spectral library was included in FastEI to improve EI-MS-based compound identification. The *in-silico* library with predicted spectra of large-scale molecules can extend the chemical space and immensely increase the coverage compared to experimental libraries. By increasing the number of compounds in the *in-silico* library to more than two million in FastEI, the false negative rate can be significantly reduced during compound identification. It can be seen from the identification results of 10 compounds in the extra test set. Without improving the coverage, all these 10 molecules cannot be identified by matching the NIST 2017. After improving the coverage, its recall@1 is 50%, recall@10 is 70%, and recall@20 is 90% by FastEI, respectively.

Secondly, the high accuracy of FastEI depends on the Word2vec spectral embedding. The mass spectra contain structural information. However, it cannot be entirely extracted by the cosine similarity based on the spectral binning vectors. Not all molecules with high structural similarity have a high cosine similarity based on spectral binning vectors. Cross-correlation can improve the correlation between spectral and structural similarity by shifting the fragment peaks in mass spectra[28]. On the other hand, it is possible to transform the mass spectra appropriately so that the cosine similarity between the transformed spectra is more relevant to the structural similarity. Inspired by the Word2vec technique in natural language processing[29], the Spec2Vec method has been developed for tandem mass spectra (MS/MS). It can learn the co-occurrence of molecular fragments in large-scale spectra and represent highly related fragments by vectors in similar directions[30]. As shown in Supplementary Fig. 4 and Fig. 5, the Word2vec model successfully mines important features that are efficient for structure "discrimination" and improves the accuracy of true/false candidate selection. There are 1413 target molecules in the test set ranked top1 by FastEI but ranked outside of the top1 by WCS. After Word2vec spectral embeddings, the same molecules have high cosine similarity, and different molecules have low cosine similarity. The possible explanation for these performance improvements is that the conversion of spectral bins to embeddings by Word2vec successfully re-scales the chemical space. At the same time, the model removes noisy bins that may interfere with the possible discrimination (i.e., noisy bins weaken the power of the discrimination). Therefore, the Word2vec spectral embedding improves the accuracy of compound identification.

Thirdly, the high speed of FastEI depends on the HNSW-based spectrum matching. The HNSW provides an efficient way to find the approximate nearest spectral embeddings of a query spectral embedding. Because of the hierarchical graph structure and the efficient graph traversal method of HNSW, the number of spectra to be matched is reduced by several orders of magnitude compared to traditional methods.

Lastly, FastEI was packaged as a standalone and user-friendly software for users without programming backgrounds. The users can quickly and accurately identify the compound by simply loading raw measured mass spectra in FastEI.

Overall, FastEI has shown excellent accuracy and speed for spectrum matching. Furthermore, we believe that FastEI can be extended to other instruments that require spectrum matching against large spectral libraries, such as tandem mass spectrometry, nuclear magnetic resonance spectroscopy, infrared spectroscopy, and Raman

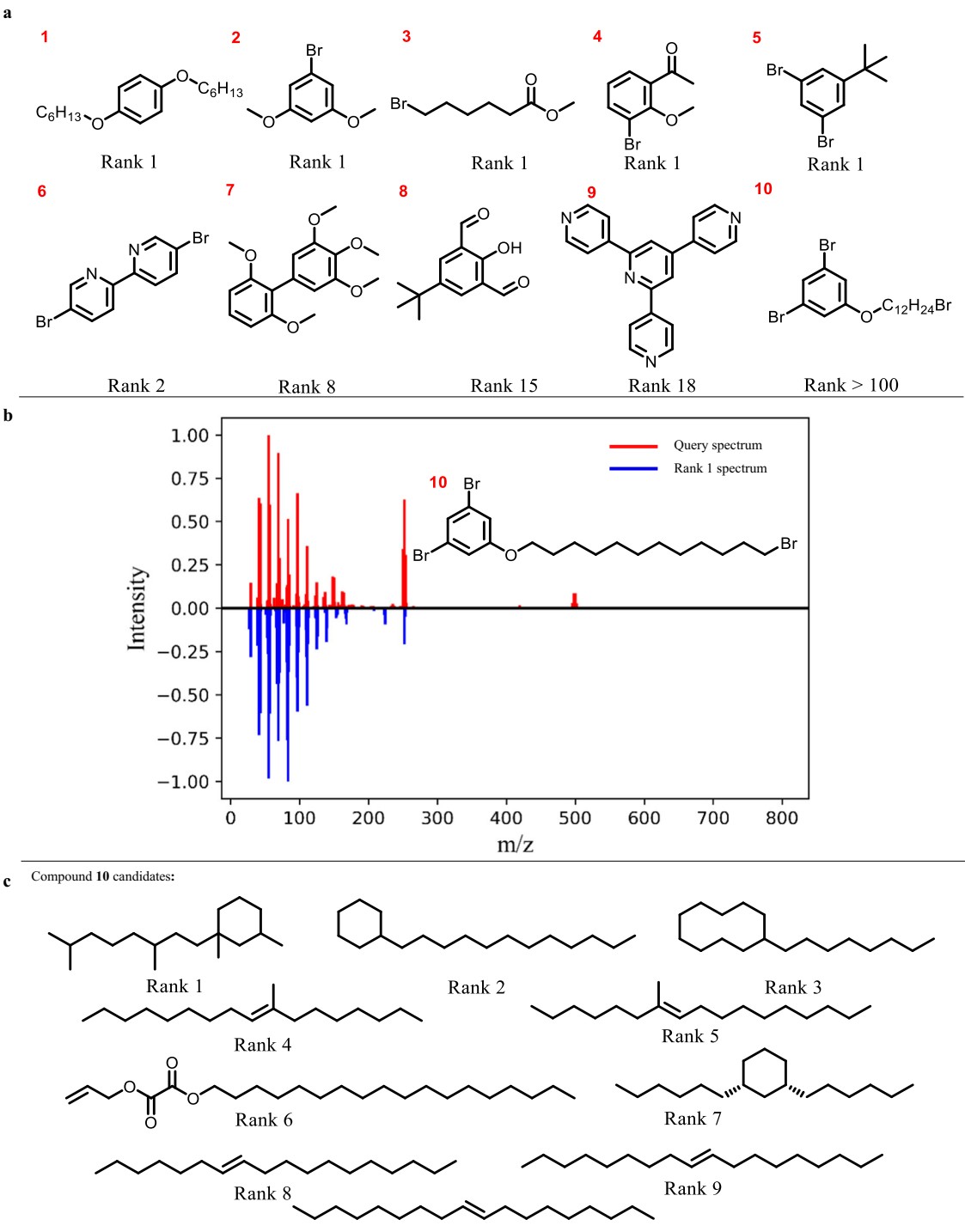

**Fig. 3 | The performance of FastEI on compounds beyond NIST 2017. a** The molecular structures of 10 compounds outside the NIST 2017 library, and their identification results by FastEI. Rank 1 means the target molecule is ranked first in the candidate list. **b** The red spectrum represents the query spectrum of compound 10. The blue spectrum represents the spectrum of the molecule ranked first in the candidate list. Rank 1 means the target molecule is ranked first in the candidate list. **c** Molecular Structure display of the top ten candidates of target compound **10**.

spectroscopy. In summary, FastEI is an integrated and user-friendly tool with GUI for ultra-fast and accurate compound identification.

## Methods

### Datasets for building the libraries

Four datasets were used to build the *in-silico* and the expanded libraries. They are NIST 2017, ChEMBL 28, HMDB 5.0[31], and the extra test set. Since the GC-MS technique is only suitable for analyzing

volatile small molecules, five filtering steps (Supplementary Fig. 6) are applied to the molecules in these four datasets to obtain reasonable datasets. They are: (1) molecular mass is less than 1000 Da; (2) molecules only contain 11 common elements H, C, O, N, P, S, Cl, F, Br, I, Si; (3) molecules are not ionic compounds; (4) molecular LogP is within the range from −12 to 24; (5) repeating molecules was deduplicated within the dataset. After filtering, the deduplication was strictly conducted among all four datasets to ensure the uniqueness

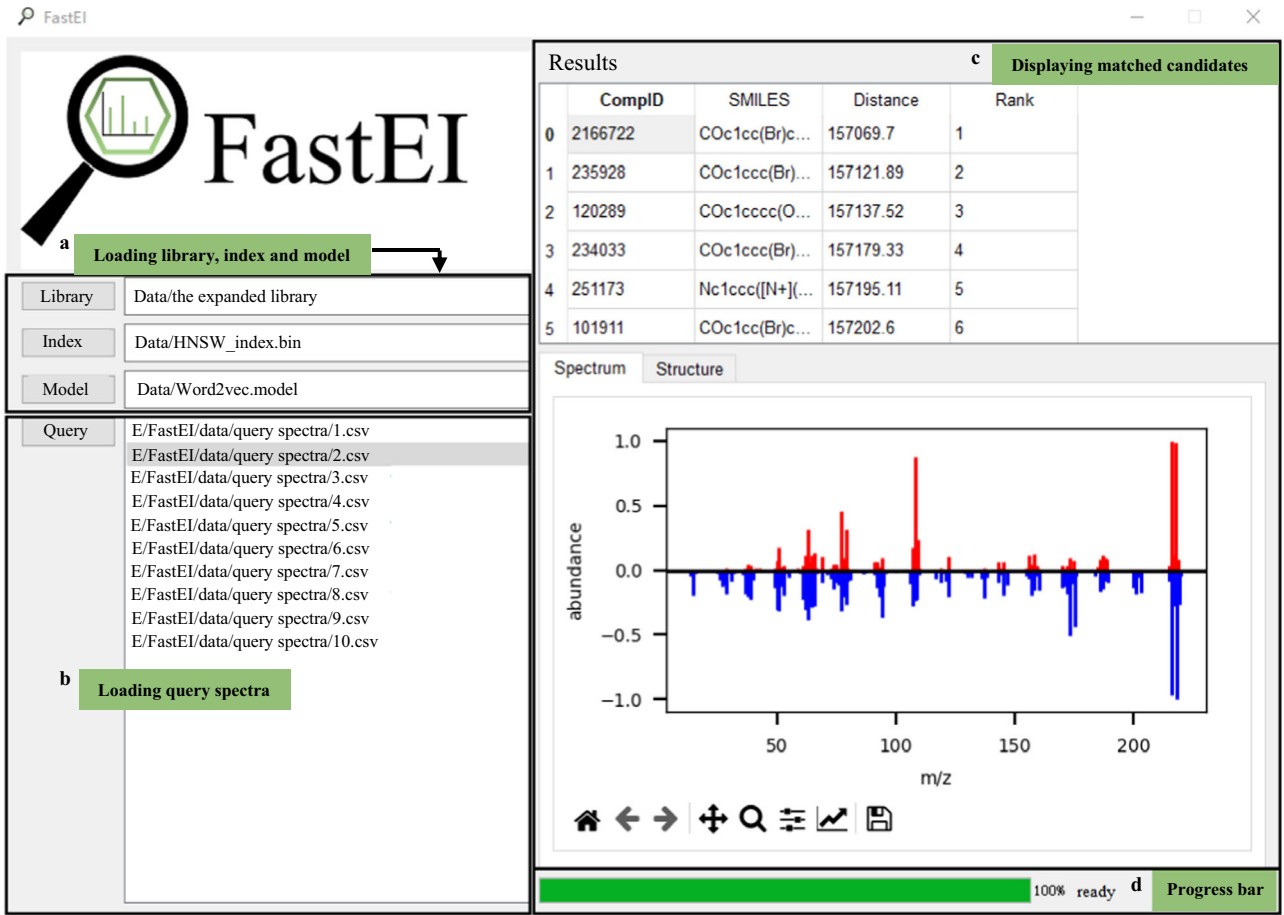

**Fig. 4 | Screenshot of FastEI. a** loading library, index, and model. **b** loading query spectra. **c** displaying the matched candidates **d** a progress bar showing the matching progress.

of the molecules in the library. The detailed information on excluded molecules by data filtering and deduplicating is displayed in Supplementary Data 1.

(a) NIST 2017. The SMILES strings of the NIST 2017 were downloaded from the NEIMS GitHub repository (https://github.com/brain-research/deep-molecular-massspec/tree/main/training_splits). It consists of three datasets, the NEIMS_training set (237,108), the NEIMS_ validation set (11,499), and the NEIMS_test set (11,600) set[8]. Using the five filtering rules (Supplementary Fig. 6), these three datasets were filtered to get the training set (232,826), validation set (11,496), and test set (11,499). The *f*-NIST (255,821) was obtained by combining the training set, the validation set, and the test set.

(b) ChEMBL 28. The gzipped structure-data file (SDF) of ChEMBL 28 was downloaded from https://ftp.ebi.ac.uk/pub/databases/chembl/ChEMBLdb/releases/chembl_28/chembl_28.sdf.gz (accessed on February 6th, 2022). The SMILES strings of molecules were read by RDKit (v2022.03.3). There are 2,066,376 molecules in the SDF file with mol files, of which 2,066,374 can be read using RDKit. After filtering with five rules and deduplicating with *f*-NIST, the number of remaining molecules is 1,890,869. The resulting library is called *f*-ChEMBL.

(c) HMDB 5.0. The zip archive of the SDF file of HMDB 5.0 was downloaded from its official website https://hmdb.ca/system/downloads/current/structures.zip (accessed on February 23rd, 2022). The SMILES strings of 217,759 molecules were read using RDKit (v2022.03.3). After filtering and deduplication with *f*-NIST and *f*-ChEMBL, the resulting library is called *f*-HMDB with 106,516 molecules.

(d) Extra test set. All small molecules in the extra test set were obtained from the organic synthesis laboratory of Prof. Pingshan Wang. These molecules are important raw materials, intermediates, or products in organic synthesis. The Bromo-substituted compounds in the extra test set, like **2, 3, 4, 5, 6,** and **10**, play an important role as intermediates in producing adsorbent materials and pharmaceuticals[32]. The ChEMBL includes many drug-like bioactive compounds, in which 4% of the molecules contain the element bromine. Compound **3**, methyl 6-bromohexanoate, is a useful intermediate for synthesizing drug carriers[33,34] and adsorbents for dye-sensitized solar cells[35]. Compound **9** is often used to synthesize metal-organic framework (MOF), which is a famous class of coordination materials[36,37]. Other oxyaromatic compounds (**1, 7, 8**) have been proven to be promising modifying agents for nylons to improve their physico-mechanical properties[38]. These ten molecules are beyond NIST 2017 and absent from *f*-ChEMBL and *f*-HMDB. Therefore, this dataset does not need to be deduplicated.

With the above data filtering and deduplicating, there are no duplicate molecules between these four datasets (*f*-NIST, *f*-ChEMBL, *f*-HMDB, and the extra test set). All the molecular structure information in the *f*-NIST, *f*-ChEMBL, *f*-HMDB, and the extra test set are prepared as the inputs of the NEIMS model to generate the predicted spectra.

**Datasets for FastEI**
Four datasets were used to train the Word2vec model, optimize the hyperparameters of the Word2vec model, and evaluate the performance of the FastEI, respectively.

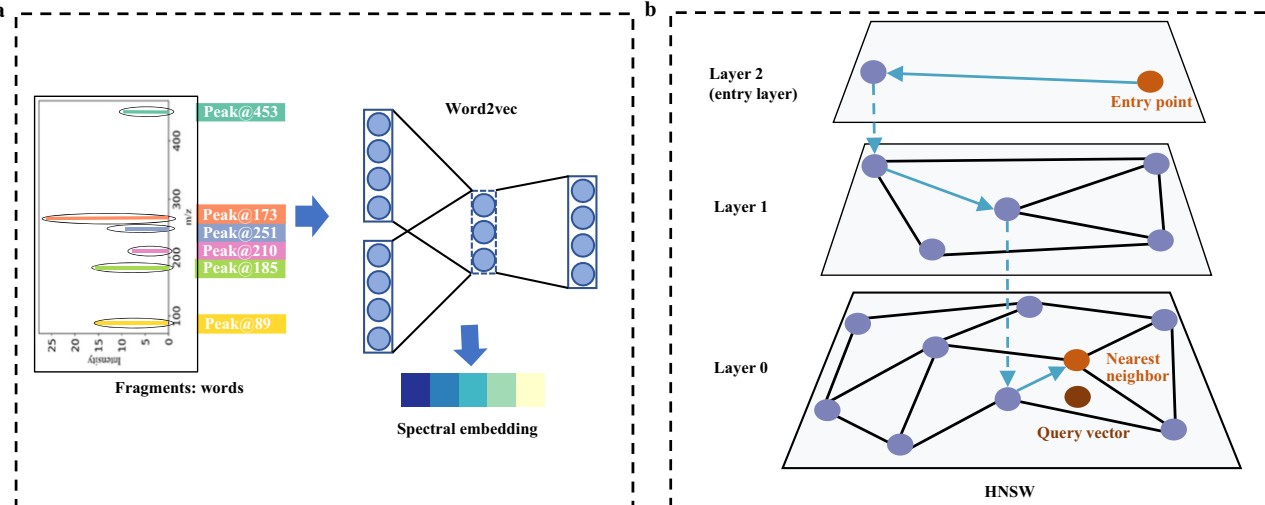

**Fig. 5 | Word2vec and HNSW. a** Transforming the spectra into low-dimensional spectral embeddings using the Word2vec model. The Word2vec model is trained on fragments in the mass spectra. Each fragment is represented by a word that contains its position up to a defined integer precision (Peak@xx). For example, a fragment at m/z 89 translates into the word "Peak@89". **b** Approximate nearest neighbor search based on the hierarchical navigable small-world graph. The top

layer is the entry point and contains only the longest links. As moving down the layers, the link lengths become shorter and more numerous. The search process is as follows: traversing edges in each layer; greedily moving to the nearest vertex until finding a local minimum in the current layer; repeating the above process until finding the nearest neighbors of the query vector in the bottom layer (*layer 0*).

(a)  Word2vec training set. The predicted spectra of molecules in the training set (232,826) and *f*-ChEMBL (1,890,869) were merged into the Word2vec training set (Supplementary Fig. 1c). It consists of 2,123,695 predicted spectra.

(b)  Validation set. The Word2vec is a self-supervised learning method and cannot be validated and tested directly. The measured spectra of the validation set were collected from the NIST 2017 library. It consists of 11,496 measured spectra. It was used to optimize the hyperparameters of the Word2vec model according to its matched results of FastEI.

(c)  Test set. The measured spectra of the test set were collected from the NIST 2017 library. It consists of 11,499 measured spectra. It was used to evaluate the performance of FastEI.

(d)  Extra test set. The experimental spectra of the extra test set were measured on a Shimadzu GC-2010 gas chromatography coupled with a Shimadzu QP2010Ultra mass spectrometer (Shimadzu, Japan), equipped with an autosampler GL 221-34618. It consists of 10 measured spectra. It was used to evaluate the performance of FastEI for molecules beyond the NIST 2017 library. The detailed experimental condition can be seen in Supplementary Note 2.

### Prediction of spectra by NEIMS and construction of the *in-silico* and expanded libraries

The ECFPs of all molecules in *f*-NIST, *f*-ChEMBL, *f*-HMDB, and the extra test set were fed into the NEIMS model to obtain their predicted mass spectra. The NEIMS model was downloaded from its official repository (https://storage.googleapis.com/deep-molecular-massspec/massspec_weights/massspec_weights.zip) without re-training. The input of NEIMS is the ECFPs, and the RDKit package is used to calculate the ECFPs from SMILES strings. The output of the NEIMS model is a spectral vector representing the intensity at each m/z. As shown in Supplementary Fig. 1b, the predicted spectra of molecules in *f*-NIST and *f*-ChEMBL were used to build the *in-silico* library, which included 2,146,690 molecules and their predicted spectra. The predicted spectra of molecules in *f*-HMDB and the extra test set were added to the *in-silico* library to build the expanded library, which included 2,253,216 molecules and their predicted spectra.

### Word2vec model building and spectral embedding

Two methods were used to represent mass spectra in this study, the binning vectors and the Word2vec embeddings. For the binning vectors, mass spectra were represented as $m$-dimensional vectors representing the intensity at each m/z. For the embedding method, the Word2vec model was adapted to learn meaningful representations from mass spectra and get $d$-dimensional embeddings. By comparing Supplementary Fig. 7a, b the Word2vec embeddings are more relevant to the chemical superclasses than the spectral binning vectors.

As shown in Supplementary Fig. 1c, the Word2vec model was trained using the Word2vec training set based on gensim[39], a Python library. The Word2vec model trained on peaks in the mass spectra (Fig. 5a) differs significantly from typical natural language processing (NLP) applications in several aspects, and some critical hyperparameters of the model also differ from the default settings. All the hyperparameters of Word2vec are listed in Supplementary Table 1. First, peaks in the mass spectra have no particular order comparable to the word order in a document. Moreover, larger windows tend to capture more topic/domain information[40]. Smaller windows tend to capture more about a specific word itself. An individual peak in mass spectra is meaningless. The combination of all peaks is meaningful in mass spectra. Hence, the window size was set to 1000. There are two ways to implement word embeddings using Word2vec: skip-gram and continuous bag of words (CBOW). The latter was observed to perform better for the embedding of mass spectra. All previously mentioned hyperparameters were adjusted according to the spectrum matching performance of FastEI on the validation set. The other hyperparameters were set as followings: (1) word vector dimension = 500, (2) negative sampling (negative = 5), (3) initial learning rate = 0.025, (4) learning rate decay per epoch = 0.00025, (5) epoch = 60. Commonly, training with a large epoch can improve the results of a Word2vec model in NLP tasks. However, this rule is not suitable for the embedding of mass spectra. When the epoch exceeds 60, the performance does not increase anymore.

Then, with the optimized hyperparameters of the Word2vec model, all the predicted spectra were transformed into low-dimensional embeddings by summing the peak embedding to build the HNSW index for library matching. The measured query spectra were also transformed into low-dimensional embeddings using the

trained Word2vec model to match the library through the HNSW index.

## HNSW-based spectrum matching

HNSW is an ANNS method based on the hierarchical graph structure and the graph traversal method. HNSW indices have no learnable parameters and do not need to be trained with data. Essentially, the HNSW index is an efficient data structure (the hierarchically multi-layer graph) to store and organize data. It can be used to improve the speed of ANNS significantly. Traditional spectrum matching methods need to calculate the similarities between the query spectrum vector and all the spectral vectors in a library. They can be calculated by the Euclidean distance between two normalized vectors using faiss.IndexFlatL2[41]. The spectrum matching time of these methods is acceptable for experimental libraries with hundreds of thousands of spectra. But for *in-silico* libraries with millions of spectra, the spectrum matching time grows significantly and affects the efficiency of compound identification. The ANNS methods are good solutions to improve the spectrum matching efficiency for large libraries. They relax the exact solution of NNS by allowing a small number of errors. The ANNS methods accelerate the spectrum matching process by efficient indexing techniques, such as locality-sensitive hashing (LSH)[42,43], space partition (tree)[44,45], and graph traversal[14,46–50]. Essentially, these indexing techniques are highly efficient approaches to retrieving candidates similar to the query spectrum. Then, the squared L2 norm is calculated only between the query spectrum and these candidates. If the indexing method is good enough, the results based on these candidates will be almost as good as those based on all the spectra in the library. Here, HNSW was chosen because of its fast speed, excellent recall rate, and high-quality implementations.

Building an index and searching with the index are two necessary steps for an ANNS-based spectrum matching method. To build an HNSW index, two most important techniques, the probability skip list and the navigable small-world graphs (NSW), are commonly used. The probability skip list consists of several additional layers of linked lists built upon an original linked list. The layers to add elements are randomly selected with an exponentially decaying probability distribution. NSW is a proximity graph with both long-range and short-range links. Due to using the greedy routing algorithm, the searching time of NSW is polylogarithmic complexity. The HNSW method can be obtained by replacing the linked lists in the probability skip list with proximity graphs, and it can also be regarded as the introduction of hierarchy into NSW. The hierarchically multi-layer graph architecture of HNSW is shown in Fig. 5b. For a spectral vector s to be added into HNSW with $L + 1$ layers, its layer level $l$ is randomly selected with an exponentially decaying probability distribution. A *search-layer* algorithm is implemented to find *ef* (size of the dynamic candidate list) nearest neighbors to s. It starts at the top layer (layer $L$) and ends until reaching the local minimum of a layer. Then, it moves down to the next layer and performs a similar search procedure with the nearest neighbor of the previous layer as an entry point. This process is repeated until reaching layer $l + 1$. From layer $l$ to layer 0, the vector s will be added to each layer. More nearest neighbors can be obtained by increasing the *ef* parameter in the *search-layer* algorithm from 1 to *efConstruction*. After adding s to this layer, $M$ nearest neighbors to s will establish bidirectional links with s. The above addition process is repeated until layer 0. Then, the vector s is successfully added to HNSW. The above steps are repeated for each spectral vector in the spectral library to add them to HNSW. After adding all spectral vectors, the HNSW index of this spectral library is built. With the built index, the nearest neighbors to the query spectral vector q can be found efficiently. From layer $L$ to layer 1, each layer is searched by the *search-layer* algorithm with $ef = 1$, and the nearest neighbor in the previous layer is used as the entry point for the next layer. For layer 0, it is searched by the *search-layer* algorithm

with *ef* set by users, and $K$ nearest neighbors in the dynamic candidate list are returned as results.

With 500 elements in each spectral embedding and more than two million spectra in the *in-silico* library, the *efConstruction* was set to 600. The other parameters *(M, ef)* are determined by the results shown in Supplementary Fig. 8. The HNSW index of FastEI was created by spectral embeddings of the *in-silico* library using hnswlib (https://github.com/nmslib/hnswlib) with $M = 64$, *efConstruction = 600*, and $ef = 300$.

## Reporting summary

Further information on research design is available in the Nature Portfolio Reporting Summary linked to this article.

## Data availability

The *in-silico* spectral library, the Word2vec model, and the index file generated in this study have been deposited in the Zenodo database under accession code 7476120. Source data are provided with this paper.

## Code availability

All code of FastEI is released under the Apache 2.0 license at https://github.com/Qiong-Yang/FastEI. This repository includes detailed instructions on how to install and run FastEI[51].

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

## Acknowledgements

This work is financially supported by the National Natural Science Foundation of China (21873116, 22273120, and 21675174) to H.L.. We are grateful for resources from the High-Performance Computing Center of Central South University.

## Author contributions

Q.Y., H.J., H.L., and Z.Z. conceived the idea and designed the database and software. Z.X., Y.L., P.W., and J.S. performed the sample preparation, data acquisition and data processing. X.F. and H.Z. performed the

data analysis. Q.Y., Z.Z., Z.X., and H.L. tested and debugged the program. Q.Y. and H.J. contributed to part of codes. Q.Y., H.J., H.L., and Z.Z. wrote the manuscript. H.L. and Z.Z. supervised the project.

## Competing interests

The authors declare no competing interests.
