## [Peer Review File · Nature Communications]

REVIEWER COMMENTS

Reviewer #1 (Remarks to the Author):

The authors present a novel application of two algorithms for performing library matching to identify EI-MS spectra: employing Word2Vec to encode EI-MS spectra and hierarchical navigable small-world graph (HNSW) to index the encoded spectra. They present evidence that this matching algorithm is orders of magnitude faster and more accurate at matching query spectra against a spectral library of predicted spectra for millions of molecules (generated using the previously published NEIMS model). This algorithm is also packaged into a standalone software package (FastEI).

The speedup observed in matching made by applying Word2Vec + HNSW matching algorithms are impressive, and I find the technical contributions to be mostly sound, except for one comment, see below.

My main critique is that the authors should highlight what they envision to be the main use case for Fast-EI. Specifically, under which contexts would it be useful to have a large library of predicted spectra and a fast matching spectral matching algorithm for EI-MS spectra?

In the manuscript, the authors test their algorithm on one possible use case: finding EI-MS matches for 10 organic synthesis compounds not present in the NIST 17 library. While their algorithm is successful in finding matches for most of the 10 synthetic compounds within the top 10 matches, it is not clear to me that the library of millions of spectra is required for this particular application. Typically an organic chemist would have a rough idea of the compounds that they are making, or possible side products. Thus, they might verify the compound based on the spectra simply by peak identification, or by running library search on a smaller library of compounds that is likely to contain the compounds of interest.

Another application proposed by the authors is in metabolomics. I am not an expert in the area of metabolomics mass spectrometry, but based on my reading of this review (<https://www.nature.com/articles/s41592-021-01197-1>), Tandem MS/MS is typically used for identifying individual compounds, which use a different type of ionization strategy (HCD or CID, rather than EI). Perhaps as a proxy indication the usefulness of FastEI for identifying metabolite spectra, the authors could apply FastEI for identifying EI-MS spectra for metabolite molecules.

In my opinion, this work represents a useful approach to efficiently searching spectral libraries that contains millions of (predicted) spectra. The greatest potential of this approach is its application to other spectrometry types, as noted in the conclusion by the authors. However, I am unsure about how useful

this approach is for querying EI-MS spectra in particular. Highlighting the need for millions-scale EI-MS coverage in spectral libraries would be helpful in demonstrating the direct impact of FastEI itself.

Additional comments:

1) NEIMS was trained to predict spectra of maximum m/z ratio of 1000 Da (https://github.com/brain-research/deep-molecular-massspec/blob/main/spectra_predictor.py#L23). If you use the NEIMS model with the provided weights linked from the github repository, as currently indicated by the methods section, the spectral predictions will all have 1000 bins (max m/z = 1000 Da). This model would have errors in predicting spectra for molecules that have mass > 1000 Da; based on the manuscript, molecules with mass up to 1500 Da are included in the library and predictions are generated for all these molecules. Additionally, I would expect library matching performed with cosine similarity ranking on NEIMS predicted spectra to have lower performance due to the missing values for m/z > 1000 Da.

2) Can you comment on the neighborhoods/groupings that are found using your algorithm? Do they correspond to specific functional group types or types of compounds? Perhaps you could compare the groupings of spectra against the ClassyFire labels for the molecules.

3) Figure 2a) For the UMAP, what is the underlying representation of the molecules. I see from the main text that you ECFP, but it would be helpful to say so in the caption.

Reviewer #2 (Remarks to the Author):

This study focused on expanding the coverage of spectral libraries and developing an accurate spectrum matching method. However, a currently reported method (ACS Cent. Sci. 5, 700–708 (2019)) was used by the authors to expand the spectral library, and the actual contribution of this paper is only to propose a method for spectrum matching.

As mentioned in the text, number of the methods have been reported for spectrum matching research, including the most commonly used WCS method (J. Am. Soc. Mass Spectrom. 5, 859–866 (1994)), and method based on neural network models (Anal. Chem. 92, 11818–11825 (2020), IEEE Trans. Pattern Anal. Mach. Intell. 42, 824–836 (2020)). To sum up, although the work does have merit, the findings do not reach the level of novelty and/or broad significance that required for publication in Nature Communications.

Reviewer #3 (Remarks to the Author):

Please see attached.

Responses to reviewer comments

Reviewer #1:

The authors present a novel application of two algorithms for performing library matching to identify EI-MS spectra: employing Word2vec to encode EI-MS spectra and hierarchical navigable small-world graph (HNSW) to index the encoded spectra. They present evidence that this matching algorithm is orders of magnitude faster and more accurate at matching query spectra against a spectral library of predicted spectra for millions of molecules (generated using the previously published NEIMS model). This algorithm is also packaged into a standalone software package (FastEI).

The speedup observed in matching made by applying Word2vec + HNSW matching algorithms are impressive, and I find the technical contributions to be mostly sound, except for one comment, see below.

Comment 1. My main critique is that the authors should highlight what they envision to be the main use case for Fast-EI. Specifically, under which contexts would it be useful to have a large library of predicted spectra and a fast matching spectral matching algorithm for EI-MS spectra?

Response: Thanks for the professional comment from reviewer #1. We have modified the introduction to highlight some main application fields of FastEI, which are life sciences fields such as metabolomics and proteomics, and organic synthesis in chemistry. We have added the use cases of FastEI in the introduction of the revised manuscript. The added sentences were excerpted here for the convenience of reviewers:

“Mass spectrometry (MS) is a convenient, highly sensitive, and reliable method for the analysis of complex mixtures, which is vital for life sciences fields such as metabolomics and proteomics, and organic synthesis in chemistry¹. High throughput identification of compounds in these complex samples is enabled by MS spectrometry generation of thousands to millions of spectra, from which many thousands of compounds are typically identified by matching the experimental MS spectra against databases containing a list of known molecular masses and fragmentation patterns. However, the vast majority of compounds in an MS experiment cannot be identified due to limited coverage in existing databases.” (Line 21-Line 35, Page 1)

For metabolomics, GC-MS is a common tool to analyze volatile, thermally stable metabolites. We have explained that the identification of metabolites requires an EI-MS library with millions of compounds by analyzing the number of volatile compounds, trends in the growth of the number of EI-MS spectra in the NIST library, and the different costs of false-positive and false-negative errors in compound identification. For details, please refer to the response of **Comment 3 of reviewer #1** and **Comment 4 of reviewer #1**.

For organic synthesis, large libraries and fast matching algorithms are needed because of the existence of numerous organic reactions (*J. Am. Chem. Soc.*, 2021, 143, 18820-18826), side products (*Org. Process Res. Dev.*, 2012, 16, 1877-1877), and novel organic reaction (*Nature* 2022, 605, 477-482; *Angew. Chem. Int. Ed.*, 2022, 61, e202204378). The high coverage of a spectral library is promising for searching for new reactivity from a larger chemical space combined with an organic autonomous synthesis robot (*Nature*, 2018, 559, 377-381). In all, organic research has the potential to generate new, unanticipated compounds, which need large libraries and fast matching algorithms for compound identification. On the other hand, we have provided a Jupyter notebook with standard operating procedures to build a small library by a given organic chemist. Organic chemists can easily switch between different libraries in FastEI according to their needs. For details, please refer to the response of **Comment 2 of reviewer #1**.

Comment 2. In the manuscript, the authors test their algorithm on one possible use case: finding EI-MS matches for 10 organic synthesis compounds not present in the NIST 17 library. While their algorithm is successful in finding matches for most of the 10 synthetic compounds within the top 10 matches, it is not clear to me that the library of millions of spectra is required for this particular application. Typically an organic chemist would have a rough idea of the compounds that they are making, or possible side products. Thus, they might verify the compound based on the spectra simply by peak identification, or by running library search on a smaller library of compounds that is likely to contain the compounds of interest.

Response: We agree with reviewer #1 that the requirement of a large library with millions of spectra should be elucidated. A small library containing the compounds of interest is suitable for a **given** organic chemist. With a large library, the spectral matching software is much more **universal** and can be used in different scenarios. With a small library, the spectral matching software is more **accurate** in recall@1 and recall@10 for a particular application. Meanwhile, large and small libraries are **compatible** with FastEI. In FastEI, one can switch to different libraries with a few mouse clicks according to their demand. In the following paragraphs, we will describe the necessity of large libraries, the accuracy of small libraries, and the compatibility of different libraries in detail.

(a). **Necessity of large libraries.** From the perspective of software developers, we expect FastEI is a universal and useful tool that can be used by researchers in organic synthesis, metabolomics, environment, pharmaceuticals, and other fields. Specifically, in organic synthesis, the library needs good molecular coverage to guarantee its universality and ensure a broad audience. In the Open Reaction Database (*J. Am. Chem. Soc.*, 2021, 143, 18820-18826), there are 2,268,124 chemical reactions. The total number of compounds, including reactants and products, is enormous. Furthermore, there are unanticipated side products (*Org. Process Res. Dev.*, 2012, 16, 1877-1877), novel organic reactions (*Nature* 2022, 605, 477-482; *Angew. Chem. Int. Ed.*, 2022, 61, e202204378). With the development of artificial intelligence (AI), automatic synthesis robots are receiving more and more attention. The synthesis robot can perform chemical reactions and analysis faster than they can be performed manually, as well as predict the reactivity of possible reagent combinations (*Nature* 2018, 559, 377-381). The high coverage of the spectral library is promising for searching for new reactivity from a larger chemical space combined with an organic autonomous synthesis robot. From the above analysis, it can be concluded that a large library with millions of compounds is only the basic requirement of an universal software for spectral matching in organic synthesis. For example, we have exhibited 10 synthesized compounds in the original manuscript. We applied NEIMS to predict the *in-silico* spectra of all possible molecules extracted from the templates of known organic reactions (*ACS Cent. Sci.* 2017, 3, 12, 1237-1245). As a result, 128,172 compounds were obtained within two reaction steps. These 10 compounds are only a small fraction (less than 1%) of all the compounds in the organic synthesis laboratory of Prof. Pingshan Wang. By analogy, the variety of compounds that may appear in the reaction system of Prof. Wang's lab is in the millions. Therefore, a large library is necessary for accurate compound identification in the organic reaction system. It will not only be helpful to identify the target compound from the mixed system of products, but also beneficial to find unexpected compounds.

(b). **Accuracy of small libraries.** For a **given** organic chemist, the reactants, reactions, products, and side products are mostly clear during his research. Using this prior information, smaller libraries can be built to improve the accuracy of compound identification further. However, we do not know which organic chemist would use FastEI to identify the compounds. Therefore, it is not possible to provide small pre-built libraries for specific applications and chemists. After thorough consideration, a feasible solution is providing the functionality for researchers to create their libraries. First, the schema of libraries of FastEI have been uploaded to its GitHub repository at <https://github.com/Qiong-Yang/FastEI/tree/main/lib/schema.png>, and the schema has been explained in README at <https://github.com/Qiong-Yang/FastEI/tree/main/lib/README.md>. Second, a Jupyter notebook (<https://github.com/Qiong-Yang/FastEI/tree/main/lib/ownlib.ipynb>) has been created with standard operating procedures for building your own libraries. It consists of preparing the required information of compounds, predicting the EI-MS spectra using NEIMS, generating an SQLite library according to the schema, embedding the predicted spectra, and adding

the embeddings into an HNSW index. Here, these collected 128,172 compounds are used to build a specific small library to explore the accuracy of the small library. Using the Jupyter notebook, a small library with 128,172 compounds and its corresponding HNSW index have been built easily. We have entered the experimental mass spectra of these 10 synthetic compounds in the extra test set into FastEI to match this small library for compound identification. The identification results are shown in Fig. R1. It can be seen that 70% correct molecular structures of all compounds are ranked in the 1st position. Compared to the large *in-silico* library, this small library improves the matching recall@1 from 50 % to 70 %. However, its application field is particularly fixed.

Fig. R1 The performance of FastEI when matching 10 synthetic compounds against the small library.

(c). **Compatibility of different libraries.** Since FastEI is very flexible in software design, it can switch between different libraries even without restarting the software. For the universality, it includes the expanded library by default. Using the standard operating procedures (<https://github.com/Qiong-Yang/FastEI/tree/main/lib/ownlib.ipynb>), researchers can build their own new libraries for their research. FastEI is compatible with both the default large library and the new libraries. Fig. R2 shows how to switch from the default library to a self-built new library. According to Fig. R2a, we can change to the self-built small library by clicking the “Database” and “Index” buttons in the main window of FastEI. Then, we can navigate to the folder containing the self-built new library (corresponding SQLite file and HNSW index file) and choose them. It can be seen from Fig. R2b that FastEI has switched from the default large library to the self-built small library successfully.

Fig. R2. Switching from the default library to a self-built new library. **a** Changing libraries using the “Database” and “Index” buttons. **b** the screenshot of FastEI after switching to the self-built small library.

Comment 3. Another application proposed by the authors is in metabolomics. I am not an expert in the area of metabolomics mass spectrometry, but based on my reading of this review (<https://www.nature.com/articles/s41592-021-01197-1>), Tandem MS/MS is typically used for identifying individual compounds, which use a different type of ionization strategy (HCD or CID, rather than EI). Perhaps as a proxy indication the usefulness of FastEI for identifying metabolite spectra, the authors could apply FastEI for identifying EI-MS spectra for metabolite molecules.

Response: Thanks to reviewer #1 for recommending this excellent review, and it provides a guide for annotation, quantification and reporting of mass spectrometry-based metabolomics. After a careful reading of this review, some interesting points have been found as follows:

(a). “We have previously estimated that upwards of **1 million different metabolites** occur across the tree of life.” (Paragraph 2, Page 747 of the review)

(b). “Consequently, many different extraction techniques and combinations of analytical methods have been developed in an attempt to achieve adequate **metabolite coverage**.” (Paragraph 1, Page 748 of the review)

(c). “Chromatography separation -> **Gas chromatography** Low-molecular-weight compounds; Mass spectrometry->Ionization source, ionization type (ESI, **EI**, APCI, etc.)” (Fig. 2, Page 752 of the review)

(d). “The orthogonal use of **chromatography (either gas or liquid based) with MS** and in some cases also tandem MS (MS/MS) fragmentation patterns provides great specificity.” (Paragraph 3, Page 752 of the review)

From the above points, it is clear that the number of metabolites in the organism is on the scale of millions. Currently, researchers are working on methods and instruments to improve the coverage of metabolites. Gas chromatography coupled with electron-ionization mass spectrometry (GC/EI-MS) is an ideal analytical method for volatile, thermally stable metabolites. Statistics of records on the *Web of Science* by searching the keyword of “GC-MS & metabolites” is shown in Fig. R3. Since 2018, the number of publications is higher than 1000 per year. It can be seen that GC-MS is widely used in metabolomics. As suggested by reviewer #1, this review supports “metabolomics as a use case of FastEI”. Meanwhile, there are 2,715 molecules in the test set that also exist in the Human Metabolome Database (HMDB). The HMDB is a freely available electronic database containing detailed information about small molecule metabolites in the human body. The recall@1 and recall@10 of FastEI of these 2,715 molecules are 34.3% and 76.9%, respectively.

Fig. R3. Statistics of records on the *Web of Science* by searching the keywords of “GC-MS & metabolites”.

Comment 4. In my opinion, this work represents a useful approach to efficiently searching spectral libraries that contains millions of (predicted) spectra. The greatest potential of this approach is its application to other spectrometry types, as noted in the conclusion by the authors. However, I am unsure about how useful this approach is for querying EI-MS spectra in particular. Highlighting the need for millions-scale EI-MS coverage in spectral libraries would be helpful in demonstrating the direct impact of FastEI itself.

Response: We strongly agree that the need for millions-scale molecular coverage in EI-MS spectral libraries should be highlighted. We will respond to this comment in terms of the necessities of high coverage and potential applications in other instruments.

(a). Necessities of high coverage. The ZINC 15 database (*J. Chem. Inf. Model.*, 2015, 55, 2324-2337) is a common database used in cheminformatics. It is a free database of **commercially-available** compounds. To date (December 19, 2022), there are 997 million compounds in this database. It provides an interesting “tranches” function to screen molecules using their molecular weights and LogP values. On its website (<https://zinc15.docking.org/tranches/home/>), one can unselect or select molecules of specific mass ranges and specific LogP ranges, but its ranges are rough. After selecting non-polar ($2 \leq \text{LogP} \leq 5$) and small molecular weight ($\text{MW} \leq 500$ Da) compounds, there are 849 million compounds left. The details are shown in Fig. R4. EI-MS is an ideal analytical method for volatile, thermally stable compounds. **Assuming that only 1% of them are thermally stable and volatile, there are still 8.49 million compounds.** From the perspective of the number of volatile compounds, it is necessary to construct million-level EI-MS spectral libraries.

The NIST EI-MS library is the most commonly used library for identifying unknown volatile compounds. It is updated every 3 years. We have obtained the number of unique compounds in different versions: NIST 02 (147,198), NIST 05 (163,198), NIST 08 (192,108), NIST 11 (212,961), NIST 14 (242,466), NIST 17 (267,376), and NIST 20 (306,869). In each new release, 20,000-40,000 compounds are added to the NIST EI-MS library. **It shows that users need identification software with a large library for better molecular coverage.** However, obtaining standard compounds and acquiring their mass spectra are costly and inefficient. In total, 159,671 molecules were added to the NIST EI-MS library between 2002 and 2020, an average of 8,871 molecules per year. Expanding the NIST EI-MS library to one million at this rate would take 78 years. Therefore, a feasible and efficient way to expand the mass spectral library is machine learning-based spectra prediction. It uses machine learning methods to learn mass spectral prediction rules from the NIST EI-MS library and then to predict the mass spectra of molecules in the molecular structure database. In FastEI, we chose the NEIMS method to generate a million-scale *in-silico* EI-MS library.

Both NIST and FastEI are endeavoring to improve the coverage of their libraries. The fundamental reason is to avoid false negatives when identifying unknown compounds. If compounds are not in the library, their identification results are all false negatives. A library should be as large as possible to improve its coverage. However, large libraries may cause problems with false positives. It can be seen that the size of the library determines the false negative rate and false positive rate of compound identification. A large library results in a low false negative rate and a high false positive rate; a small library results in a high false negative rate and a low false positive rate. Essentially, the size of the library is associated with **the balance between the false negative rate and the false positive rate.** In compound identification, there are no remedies for false negatives. For false positives, there are many remedies, and information on dimensions such as molecular weight (acquired by GC-CI-Orbitrap-MS*), retention index, and domain knowledge can be used for effective filtering. Therefore, it can be said that **the cost of false negatives is much higher than false positives when identifying compounds.** By increasing the number of compounds in the library to more than two million in FastEI, the false negative rate can be significantly reduced during compound identification. It can be seen from the identification results of 10 molecules in the extra test set. Without improving the coverage, all these 10 molecules cannot be identified, and the recall@x is 0%. After improving the coverage, its recall@1 is 50%, recall@10 is 70%, and recall@20 is 90%.

(b). Potential applications in other instruments. In addition to EI-MS, the methods in FastEI are suitable for matching tandem mass spectra, nuclear magnetic resonance (NMR) spectra, infrared spectroscopy, Raman spectroscopy, etc. Here, we use tandem mass spectra and NMR spectra as

* <https://assets.thermofisher.com/TFS-Assets/CMD/brochures/BR-10445-GC-MS-Q-Exactive-Orbitrap-BR10445-EN.pdf>

examples to illustrate in detail.

For tandem mass spectra, complex samples are separated by high-performance liquid chromatography (HPLC), the mass spectra are obtained by electrospray ionization (ESI), and tandem mass spectra (MS/MS) are obtained by collision-induced dissociation (CID). Because more than 85% of natural chemical compounds are polar and thermally labile, liquid chromatography-mass spectrometry (LC-MS) has much better coverage of a wide range of chemicals than GC-MS. More severely, only 30,999 compounds have tandem mass spectra in NIST 2020. Therefore, the coverage problem of tandem mass spectra libraries is even more severe than the EI-MS libraries. The Word2vec model can solve the sparsity in MS/MS spectra, and HNSW can efficiently match million or 10 million MS/MS mass spectra effectively. However, the **existing prediction methods for tandem mass spectra are insufficient in terms of accuracy, generality, and efficiency**. When predicting the mass spectra of molecules in molecular structure libraries on a large scale, there are problems, such as poor quality of predicted tandem mass spectra, some molecules not in the application domain, and time-consuming prediction. If a large number of tandem mass spectra are combined with deep learning to train an accurate, efficient model with good generalization performance, FastEI should work well for matching the *in-silico* library of tandem mass spectra.

For NMR spectra, the most common types are proton and carbon-13 NMR spectra, which identify the hydrogen atoms and carbon atoms in organic molecules to determine their structures. It is an unbiased method for the detection of any molecules that contain carbon or hydrogen. Organic compounds are generally defined as compounds that contain carbon-hydrogen or carbon-hydrogen bonds. NMR spectroscopy can detect mostly all organic compounds. Recently, deep learning-based models can predict the chemical shifts in NMR matching DFT-level performance (*Magn. Reson. Chem., 2022, 60, 1021-1031*). Therefore, FastEI is very suitable to match the NMR spectra. However, NMR is less sensitive than mass spectrometry and cannot separate complex samples effectively. **It often analyzes pre-separated samples with reasonably pure compounds**. We believe that NMR will be expanded to identify components in complex mixtures with the advancement of LC-NMR hyphenation technology.

		Molecular Weight (up to, Daltons)											Totals, by LogP
		200	250	300	325	350	375	400	425	450	500	>500	
LogP (up to)	-1	10,000	100,000	1,000,000	1,000,000	1,000,000	1,000,000	1,000,000	1,000,000	1,000,000	1,000,000	1,000,000	0
	0	10,000	1,000,000	1,000,000	1,000,000	1,000,000	1,000,000	1,000,000	1,000,000	1,000,000	1,000,000	1,000,000	0
	1	10,000	1,000,000	1,000,000	1,000,000	1,000,000	1,000,000	1,000,000	1,000,000	1,000,000	1,000,000	1,000,000	0
	2	497,750	5,391,816	25,622,912	32,914,848	67,733,100	28,989,280	19,267,814	10,563,987	9,090,177	8,721,010	70,000	208,702,694
	3	189,326	2,643,678	14,831,118	19,486,349	40,600,593	20,281,809	15,126,147	9,325,848	8,120,159	7,879,918	70,000	138,484,945
	3.5	108,266	2,075,334	13,281,388	18,060,096	37,030,641	22,002,857	17,838,728	12,045,788	10,699,251	10,674,949	70,000	143,814,298
	4	48,705	1,336,320	10,135,959	14,349,999	29,671,752	21,055,698	18,737,428	13,954,286	12,511,433	12,736,846	70,000	134,538,426
	4.5	15,100	613,109	6,131,454	8,128,568	12,531,547	15,472,307	16,892,846	14,129,429	12,864,529	13,378,049	70,000	100,156,938
	5	1,993	170,043	2,873,064	4,632,339	7,889,352	10,959,424	12,773,295	12,356,636	11,562,431	12,208,182	115,000	75,426,759
	5.5	94	21,765	852,691	1,919,530	3,985,092	6,416,455	8,397,678	9,021,197	8,818,548	9,321,913	100,000	48,754,963
	6	0	0	0	0	0	0	0	0	0	0	0	0
Totals, by Weight		861,234	12,252,065	73,728,586	99,491,729	199,442,077	125,177,830	109,033,936	81,397,171	73,573,528	74,920,867	0	849,879,023 Substances 1K Tranches

Fig. R4. The “tranches” function enables to screen molecules using their molecular weights and LogP values

Additional comments:

Comment 5. NEIMS was trained to predict spectra of maximum m/z ratio of 1000 Da (https://github.com/brain-research/deep-molecular-masspec/blob/main/spectra_predictor.py#L23). If you use the NEIMS model with the provided weights linked from the github repository, as currently indicated by the methods section, the spectral predictions will all have 1000 bins (max m/z = 1000 Da). This model would have errors in predicting spectra for molecules that have mass > 1000 Da; based on the manuscript, molecules with mass up to 1500 Da are included in the library and predictions are generated for all these molecules. Additionally, I would expect library matching performed with cosine similarity ranking on NEIMS predicted spectra to have lower performance

due to the missing values for $m/z > 1000$ Da.

Response: Thanks for pointing out the applicable range of molecular weights of the NEIMS model. We use the weights of the NEIMS model downloaded from https://storage.googleapis.com/deep-molecular-massspec/massspec_weights/massspec_weights.zip. According to the hyperparameter defined in the `spectra_predictor` module (`"max_mass_spec_peak_loc": 1000`)[†], the NEIMS model only applies to compounds with molecular weights less than or equal to 1000 Da. Therefore, we have counted molecules with molecular weights greater than 1000 Da in the datasets and libraries. In the mainlib of NIST 2017, there are 37 molecules (0.01%) with molecular weights greater than 1000 Da. In the in-silico library, there are 20,031 molecules (0.9 %) with molecular weights greater than 1000 Da. There are no molecules with molecular weights greater than 1000 Da in the test set and the extra test set. The experimental mass spectra of these 37 molecules (mass > 1000 Da in NIST 2017) are compared with their NEIMS-predicted mass spectra. The experimental mass spectra of 37 randomly selected molecules (mass < 1000 Da in NIST 2017) from the test set are compared with their NEIMS-predicted mass spectra. The weighted cosine similarities between the experimental and predicted mass spectra of these 74 molecules are shown in Fig. R5a, and it can be seen that the weighted cosine similarities of 37 molecules (mass > 1000 Da) are significantly lower than 37 molecules with molecular weights below 1000 Da. Meanwhile, the predicted mass spectra of molecules with molecular weights greater than 1000 Da are not reasonable, as seen from the experimental versus predicted mass spectra plot in Fig. R5b. **It is not a problem of the NEIMS model. There are only 37 molecules with molecular weights greater than 1000 Da in the mainlib of NIST 2017. It is challenging to train NEIMS models that can predict molecular mass spectra (mass > 1000 Da). All models have their application domains. One must ensure that the predicted data is within their application domains.** For this reason, we set the molecular weight filter to 1000 Da and removed the molecules above 1000 Da from the datasets. The Word2vec model was retrained using the new spectra. The in-silico library and HNSW index in the FastEI software were also updated. We recalculated the matching performance of the mainlib test set on the database after the updating. The results are listed in Table R1. The overall performance of FastEI is significantly better than the WCS.

Furthermore, it is interesting to investigate the effect of the m/z range on matching performance according to the comment from reviewer #1. The existing NEIMS model can only predict mass spectra in the range of 1-1000 Da. There are 37 molecules (0.01%) with molecular weights greater than 1000 Da in the *mainlib* of NIST 2017. Due to the small number of molecules with experimental mass spectra above 1000 Da, it is also difficult to train NEIMS models that can predict mass spectra in the 1-1500 Da range. It is the reason that NEIMS sets the `max_mass_spec_peak_loc` hyperparameter to 1000 Da. As shown in Fig. R5c, the molecular weights of molecules in the test set are mainly distributed below 400 Da. To investigate the relationship between the m/z range and matching performance, we have removed 60% or 80% of the high m/z values and intensities. Thus, the original m/z range of 1-1000 Da becomes 1-400 Da and 1-200 Da, respectively. We have built 400Da and 200Da in-silico libraries, respectively. After finishing these preparations, we tested the matching performance of WCS using the test set on 1000Da, 400Da, and 200Da in-silico libraries, respectively. The matching performance is evaluated with the `recall@1` metric. The `recall@1` metrics of WCS on the 1000 Da, 400Da, and 200Da in silico libraries are 30.2%, 29.5%, and 17.9%, respectively. As the m/z range becomes narrow, the matching performance of WCS decreases. It is consistent with the expectation of reviewer #1 except for the m/z range: "Additionally, I would expect library matching performed with cosine similarity ranking on NEIMS predicted spectra to have lower performance due to the missing values for $m/z > 1000$ Da."

According to Stephen E. Stein's investigation (J. Am. Soc. Mass Spectrom., 1994, 5, 859-866), the characteristic peaks with higher m/z values in mass spectra contain richer structural information. The original sentence of the article reads, "Because the higher mass peaks in a spectrum are the most diagnostic, weighting their contribution in the algorithm also improves performance.". Therefore, there is an urgent need to develop methods and models with better extrapolation performance in subsequent research, which can predict a broader range of m/z .

[†] https://github.com/brain-research/deep-molecular-massspec/blob/main/spectra_predictor.py#L23

Fig. R5. Evaluation of NEIMS models on molecules with molecular weights greater than 1000 Da. **a** weighted cosine similarities between the experimental and predicted mass spectra of 74 molecules from the NIST 2017 library. **b** the experimental versus predicted mass spectra of typical molecules with molecular weights greater than 1000 Da. **c** the molecular weights of the molecules in the test set.

Table R1[‡]. The *mainlib* test set spectrum matching performance on the *in-silico* library.

Method	recall@1 (%)	recall@10 (%)	Run time of per spectrum matching
FastEI	36.7	80.4	0.0042
WCS	30.2	73.5	2.4849
FastEI + mass filter ^a	45.3	88.3	-
WCS + mass filter ^a	37.1	81.6	-

^aThe mass filter was set to 5 Da of the query molecule's mass

Comment 6. Can you comment on the neighborhoods/groupings that are found using your algorithm? Do they correspond to specific functional group types or types of compounds? Perhaps you could compare the groupings of spectra against the ClassyFire labels for the molecules.

Response: We really appreciate this excellent and insightful comment from reviewer #1. It is similar to comment 1 of reviewer #3 but from a different perspective. As suggested by the reviewers, we have extracted groupings from FastEI, examined the relationship between the groupings and compound types, and compared the consistency between the groupings and the ClassyFire labels. The specific methods and detailed results of each step are presented here.

(a). **Extraction of groupings from FastEI.** In the application programming interface (API) of *hnswlib*[§], there are 14 public APIs, including *init_index*, *add_items*, *mark_deleted*, *unmark_deleted*, *resize_index*, *set_ef*, *knn_query*, *load_index*, *save_index*, *set_num_threads*, *get_items*, *get_ids_list*, *get_max_elements*, and *get_current_count*. However, none can extract the groupings from the *in-silico* library directly. To address this problem, we construct an extraction method for groupings based on the HNSW index. Given a representative query molecule, its spectral embedding is retrieved from the *in-silico* library and fed into the *knn_query* function to get its 10 nearest molecules. In this way, we can get 10 neighbor molecules of each representative query molecule.

(b). **Consistency between groupings and compound types.** With the molecular classes in Fig. 2a of the manuscript, we selected 9 representative molecules and obtained their neighbors, and used them as the groupings for further analysis. The structures of 9 selected representative molecules are shown in Fig. R6a. For all the 90 molecules, their ECFPs (*J. Chem. Inf. Model.*, 2010, 50, 742-754) were calculated by RDKit. The UMAP plot of the ECFPs of molecules in groupings is shown in Fig. R6b. Molecules are clustered into 9 classes by their ECFPs. There is a good consistency between the compound types and the groupings obtained based on ECFPs.

(c). **Accuracy between groupings and ClassyFire.** A grouping has a query molecule, and the chemical taxonomy of this grouping can be set as the chemical taxonomy of its query molecule. For each grouping in the 9 groupings, the SMILES string of its query molecule was used as the input of the ClassyFire web service at <http://classyfire.wishartlab.com/#chemical-text-query>. The taxonomic classification of the grouping was downloaded as a JSON file. This JSON file was parsed using Python, and the taxonomy tree of this grouping was obtained, including kingdom, superclass, and class. After processing the 9 groupings according to the above process, the taxonomy tree of each grouping can be obtained.

Each molecule in a grouping has a SMILES string that can be used to predict its chemical taxonomy using ClassyFire. It is inefficient to submit them one by one. We have written a Python script to generate the inputs for all the molecules in a grouping. The generated inputs strictly follow the formatting requirements of ClassyFire: one entry per line containing a SMILES string preceded by an identifier with tab-delimiter. The inputs were copied and pasted to the multi-line text input field of ClassyFire, and the task was submitted by clicking the submit button. After finishing the task, we downloaded the JSON files and parsed them using Python to obtain the taxonomy trees of all the molecules in the grouping. Using the above method, the taxonomy trees were obtained for the molecules in all 9 groupings.

[‡] This is Table 1 the revised manuscript, and it is presented here for the convenience of reviewers.

[§] <https://github.com/nmslib/hnswlib#short-api-description>

Fig. R6. Analyzing the relationships between FastEI groupings and compound types. **a** 9 representative query molecules. **b** UMAP plot of the ECFPs of molecules in groupings. **c** Accuracies of the superclass and class for 9 groupings.

After obtaining the taxonomy tree of the grouping and the taxonomy tree of each molecule in the grouping, we calculated the accuracy between groupings and ClassyFire. For kingdom in the taxonomy tree, all groupings and the molecules within the groupings are predicted as organic compounds. For the superclass and class in the taxonomy trees, we compared the superclass and class of each molecule in a grouping with the superclass and class of its grouping, and the accuracy was calculated in terms of the superclass and class for this grouping. The accuracies of the superclass and class for 9 groupings are shown in Fig. R6c. It can be seen that the accuracies of the superclass are in the range of 40% to 100%, and the accuracies of the class are in the range of 30% to 100%. Therefore, the ClassyFire labels of molecules in the grouping of FastEI have some consistency when compared with the ClassyFire labels of the query molecule of the grouping.

Comment 7. Figure 2a) For the UMAP, what is the underlying representation of the molecules. I see from the main text that you ECFP, but it would be helpful to say so in the caption.

Response: Yes, the underlying representation of molecules for the UMAP plot in the original Fig. 2a is ECFPs. In Python, they can be calculated by invoking the `GetMorganFingerprintAsBitVect` function in the `AllChem` module of RDKit. It can be seen that ECFPs are called Morgan fingerprints** in RDKit because the core idea of ECFPs comes from the Morgan algorithm (*J. Chem. Doc.*, 1965, 5, 107-113). We have added the ECFPs in the caption of Fig. 2b in the revised manuscript, and the revised caption is as follows:

“Fig. 2 Information of datasets and performance of FastEI. a The molecule classes predicted by ClassyFire for the test set. b The visualization of the ECFPs of 240,000 molecules randomly selected from f-CHEMBL and 232,826 molecules from the training set by UMAP. c The spectrum matching time of FastEI and WCS on libraries with different sizes. d The contribution of Word2vec-based embeddings and HNSW to FastEI (WCS: weighted binning + cosine similarity, EC: embeddings + cosine similarity, BH: weighted binning + HNSW, FastEI: embeddings + HNSW). e The performance of FastEI and WCS on the test set in terms of recall rates at different top x levels.” (Line 2-Line 6, Page 4)

** <https://www.rdkit.org/docs/GettingStartedInPython.html#morgan-fingerprints-circular-fingerprints>
http://www.rdkit.org/docs/cppapi/namespaceRDKit_1_1MorganFingerprints.html#abf2df7ebf39c700faaa775be5e4bbb70

Reviewer #2:

Comment 1. This study focused on expanding the coverage of spectral libraries and developing an accurate spectrum matching method. However, a currently reported method (ACS Cent. Sci. 5, 700 – 708 (2019)) was used by the authors to expand the spectral library, and the actual contribution of this paper is only to propose a method for spectrum matching.

As mentioned in the text, number of the methods have been reported for spectrum matching research, including the most commonly used WCS method (J. Am. Soc. Mass Spectrom. 5, 859 – 866 (1994)), and method based on neural network models (Anal. Chem. 92, 11818 – 11825 (2020), IEEE Trans. Pattern Anal. Mach. Intell. 42, 824 – 836 (2020)). To sum up, although the work does have merit, the findings do not reach the level of novelty and/or broad significance that required for publication in Nature Communications.

Response: Thanks. We have developed the FastEI that allows users to quickly search an EI-MS spectrum in the million-scale *in-silico* library with a user-friendly GUI. As reviewer#3 said, the study is highly expected to enhance the compound identification on unknowns that are measured by EI-MS in the field of pharmaceuticals, chemical synthesis, environmental, agricultural, and industrial chemistry, and others. The reported method, NEIMS (ACS Cent. Sci. 2019, 5, 700-708), was used to generate the million-scale *in-silico* library to overcome the coverage problem of the experimental library.

Traditional spectral search methods apply weighted cosine similarity (WCS) between a query spectrum with spectra in the library. It sometimes fails to achieve accurate molecular identification, especially in cases where the experimental library does not contain target molecules. Meanwhile, the structural similarity information cannot be entirely extracted by WCS based on the spectra binning vectors because not all small molecules with high structural similarity have high WCS based on spectral binning vectors. The identification accuracy based on WCS can be improved (as shown in Table R2). Moreover, matching the million-scale *in-silico* library using WCS is time-consuming compared to the FastEI (as shown in Table R2).

Table R2^{††}. The *mainlib* test set spectrum matching performance on the *in-silico* library.

Method	recall@1 (%)	recall@10 (%)	Run time of per spectrum matching (s)
FastEI	36.7	80.4	0.0042
WCS	30.2	73.5	2.4849
FastEI + mass filter ^a	45.3	88.3	-
WCS + mass filter ^a	37.1	81.6	-

^a The mass filter was set to 5 Da of the query molecule's mass

Figure 1. Deep convolutional neural network for the EI-MS library search. Activation functions in dense layers: ReLU and identity. Loss function RankNet was used.

Fig. R7 Figure 1 of the article (Anal. Chem. 2020, 92, 11818-11825)

Matyushin et al. utilize a convolutional neural network (CNN) to search the EI-MS library and achieve better accuracy than the default method of NIST MS Search software. The primary aim of this work was to develop the deep learning ranking model based on a CNN model (deep CNN), which can rank candidates for the EI-MS library search in the NIST 2017 database. As shown in Figure 1 of the article (Anal. Chem. 2020, 92, 11818-11825), the query spectrum and its candidates'

^{††} This is Table 1 the revised manuscript, and it is presented here for the convenience of reviewers.

spectra are as input to CNN. Figure 1 of the deep CNN article is provided here as Fig. R7 for the convenience of the reviewers.

The data filtering procedure of Matyushin’s work is the exclusion of compounds with molar mass or m/z of peaks in a spectrum of more than 750 and compounds with structures that cannot be processed by Chemical Development Kit 2.3 (CDK). To get the spectrum matching performance comparison of the deep CNN and FastEI. The *in-silico* library and the *replib* test set from the FastEI were used as the “reference set” and “query set” after excluding compounds with molar mass or m/z of peaks in a spectrum of more than 750. There are 52,094 and 14 excluded spectra, and 2,094,596 and 11,485 remained in the “reference set” and “query set”, respectively. The “query set” matched the “reference set” based on the pre-trained model to get the matching performance of the *replib* test set of deep CNN, which is available at <https://www.doi.org/10.6084/m9.figshare.12685790>. The matching results of FastEI and deep CNN are shown in Table R3. For the matching speed, the run time of FastEI is 0.0042 s / query spectrum, and the run time of deep CNN is 102.2584 s / query spectrum. FastEI is about 24,347 times faster than deep CNN when matching one spectrum. For the matching accuracy, the FastEI achieves a matching result with 36.7 % recall@1 and 80.4 % recall@10. The deep CNN achieves a matching result with 33.1 % recall@1 and 76.1 % recall@10. The matching accuracy of FastEI is better than the deep CNN of 3.3 and 3.6 percentage points in recall@1 and recall@10, respectively. In short, the FastEI method outperforms the deep CNN method in matching speed and accuracy.

Table R3. The spectrum matching performance of *replib* test set of FastEI, deep CNN, and WCS.

Method	recall@1 (%)	recall@10 (%)	Run time per spectrum matching (s)
FastEI	36.7	80.4	0.0042
deep CNN	33.3	76.3	102.2584

An approach for the approximate K-nearest neighbor search based on navigable small world graphs with controllable hierarchy (Hierarchical NSW, HNSW) is reported in the article (*IEEE Trans. Pattern Anal. Mach. Intell.* 2020, 42, 824-836). As shown in Fig. R8, the Word2vec-based embeddings improve accuracy, while HNSW boosts the matching speed. The FastEI method achieves high accuracy and speed by combining Word2vec-based embeddings and HNSW.

Fig. R8^{**} Analyzing the contribution of Word2vec-based embeddings and HNSW in FastEI (WCS: binning + cosine similarity, EC: embeddings + cosine similarity, BH: binning + HNSW, FastEI: embeddings + HNSW).

^{**} This is Fig. 2d in the revised manuscript, and it is presented here for the convenience of reviewers.

In conclusion, FastEI showed the potential to overcome traditional coverage problems by applying *in-silico* library constructed by NEIMS that can predict spectra from structure databases, such as ChEMBL with structural information of millions of chemicals. This study addressed the issue of low matching accuracy and increasing search time along with library size by applying Word2vec and HNSW in the software architecture. Word2vec showed high performance in accuracy for compound identification, and HNSW contributed to speeding up the library search.

Furthermore, the broad significance and novelty of the FastEI are summarized here for the convenience of the reviewers.

(a). Broad significance. Gas chromatography-mass spectrometry (GC-MS) has long been considered one of the premiere analytical tools for small molecule analysis (*Nat. Protoc.* 2006, 1, 387-396; *Trends Anal. Chem.*, 78, 2016, 23-25). The NIST electron ionization library is the most widely used for molecular identification. Compound identification is the foundation of chemistry. Without correct compound identification, it is difficult to advance chemical research. Compound identification in GC-MS is currently achieved by comparing a query mass spectrum with reference mass spectra in a library via spectrum matching. However, this approach fails to identify molecules absent from the existing libraries. Only approximately 20% of chromatographic peaks extracted from the GC-MS dataset can be identified by spectrum matching against libraries based on similarity metrics. The bottleneck lies in the *compound coverage* of spectral libraries. There are more than 111 million molecules with structures in PubChem and two million bioactive molecules in ChEMBL. There are only 0.27 million compounds with electron ionization mass spectra in NIST 2017 library. If the sample contains compounds that are absent from spectral libraries, the correct identification of them is impossible when using spectrum matching methods. There are no remedies for false negatives. For false positives, there are many remedies, and information on dimensions such as molecular weight, retention index, and domain knowledge can be used for effective filtering. Therefore, it can be said that the cost of false negatives is much higher than false positives when identifying compounds. Generating *in-silico* spectra is an effective solution to extend the coverage of spectral libraries. The NEIMS method takes the Morgan fingerprints of molecules as inputs to predict their spectra. It enables large-scale *in-silico* spectra generation from molecular structures. The *in-silico* library with predicted spectra of large-scale molecules can extend the chemical space and immensely increase the coverage compared to experimental libraries (e.g., NIST 2017 and MassBank). However, large libraries increase the probability of false positives and increase the search time. So it is urgent to explore more accurate and faster database search methods

FastEI solves the problems of coverage, accuracy, and speed in traditional methods to a certain extent. It is highly expected to enhance the compound identification with measured EI-MS in the field of metabolomics (*Trends Anal. Chem.*, 2007, 26, 9, 855-866.), pharmaceuticals (*Nat Rev Drug Discov.* 2021, 20, 200–216), chemical synthesis (*Angew. Chem.* 2011, 123, 12448–12452), environmental (*Chem. Rev.* 2015, 115, 10, 3919–3983), agricultural (*Plant Cell Physiol.*, 2011, 53, e1) and others (*ACS Nano* 2021, 15, 1, 894–903). Furthermore, we believe that FastEI can be extended to other instruments that require spectrum matching against large spectral libraries, such as tandem mass spectrometry, nuclear magnetic resonance spectroscopy, infrared spectroscopy, and Raman spectroscopy.

(b). Novelty of FastEI. Accuracy and speed have always been two mutually constraining goals in algorithm and software development, but FastEI significantly improves accuracy, speed, coverage, and facilitation. These points definitely show its strengths and novelties.

(b1). Spectrum matching accuracy from Word2vec embedding. A Word2vec model was adapted to learn meaningful representations of mass spectra and get *d*-dimensional spectral embeddings. It can learn the co-occurrence of molecular fragments in large-scale spectra and represent highly related fragments by vectors in similar directions. The high accuracy of FastEI should be mainly attributed to Word2vec-based spectral embeddings.

(b2). Ultra-fast speed from HNSW. The HNSW method was chosen here to match the million-scale spectral library. Because of the hierarchical graph structure and the efficient graph traversal method of HNSW, the match speed is improved by several orders of magnitude compared to traditional methods. The HNSW can provide an efficient way to quickly find the approximate nearest spectral embeddings of a query spectral embedding. The high speed of FastEI should be

mainly attributed to HNSW-based spectrum matching.

(b3). High coverage of million-scale *in-silico* library. In this study, 2,253,216 predicted spectra are generated by NEIMS from molecular structures to build the *in-silico* library. This *in-silico* library significantly extends the chemical space and immensely increases the coverage compared to traditional experimental libraries (e.g., NIST 2017 and MassBank).

(b4). Facilitation of FastEI software. FastEI provides an integrated spectrum matching software with a large-scale *in-silico* library and matching algorithms. Meanwhile, according to user demand, FastEI can switch to different libraries with a few mouse clicks without restarting the software. The installation package is available at <https://github.com/Qiong-Yang/FastEI/releases>. Overall, FastEI is a convenient and easy-to-use software for quickly searching a million-scale *in-silico* library.

Reviewer #3:

This study developed FastEI that allows users to search individual EI-MS spectrum in million-scale in-silico library with user-friendly GUI. The outcome is interesting and would attract attention from wide ranges of communities. Traditional spectral search system applies cosine similarity between query spectrum with library of measured spectrum. It sometime fails to reach correct compound, especially in case that the library does not contain its measurement data. The study showed potential to overcome such traditional problem by applying in-silico library constructed by NEIMS that can predict spectrum from chemical structure and database ChEMBL which contains structural information on over millions of chemicals. Furthermore, the study addressed the issue of increasing search time along with library size by applying Word2vec and HNSW in the software architecture. Word2vec showed high performance in accuracy for compound identification, and HNSW contributed speeding up in the library search. The study is highly expected to enhance the compound identification on unknowns that measured by EI-MS in the field of pharmaceuticals, chemical synthesis, environmental, agricultural, and industrial chemistry, and others. Prior to publication, authors should address the following issues.

Major Comments

Comment 1. Is there a difference in mean matching value of highly ranked list (e.g. top 10) between FastEI and traditional methods such as WCS with in-silico library or NIST17? If the FastEI improves the average value not only recall@1 and 10, it indicates that the in-silico library approach has an advantage in suggesting rough structure (i.e. fuzzy structure) of the query rather than traditional method.

Response: Thanks for this insightful and constructive comment from reviewer #3. Yes, FastEI is better than WCS in terms of the mean matching value of the top 10 candidates when matching the *in-silico* library, and it can recommend reasonable rough structures. The detailed process and results are as follows:

(a). Mean matching value. The 11,499 experimental mass spectra of the test set exported from the mainlib were chosen as the inputs of FastEI and WCS. After spectral matching, their matching results were obtained, and the top 10 candidates for each molecule were extracted from the matching results of FastEI and WCS, respectively. Since FastEI calculates the squared L2 norm distance and WCS calculates the weighted cosine similarity, it is impossible to compare their matching values directly. The matching value of FastEI is based on spectral embeddings, and there is no m/z for calculating the weighted cosine similarity. Therefore, the cosine similarity is adopted as the matching value for comparing the top 10 candidates of FastEI and WCS. The mean cosine similarity was calculated with the spectral embeddings of each molecule and its top 10 candidates for FastEI. The mean cosine similarity was calculated with the mass spectrum of each molecule and its top 10 candidates for WCS. For 11,499 molecules in the test set, the violin plot in Fig. R9a show the mean matching values of FastEI and WCS, respectively. One can observe that the distribution of mean matching values of FastEI is slightly larger than the distribution of WCS. There are shared candidates between the top 10 candidates of FastEI and the top 10 candidates of WCS for many molecules. After **removing these shared candidates**, the violin plot of the mean matching values of FastEI and WCS is redrawn in Fig. R9b. It can be seen that the mean matching values of unique candidates of FastEI is significantly larger than the mean matching values of unique candidates of WCS. Therefore, FastEI is better than WCS in terms of the mean matching value of the top 10 candidates when matching the *in-silico* library. The advantage of FastEI over the WCS method in terms of mean matching value is more significant when considering their unique candidates.

(b). Rough structure. To evaluate the quality of the rough structures (top 10 candidates) recommended by FastEI and WCS, it is necessary to define a suitable criterion. The quality of a rough structure is proportional to the number of shared substructures between its query molecule. The ECFPs, (*J. Chem. Inf. Model.*, 2010, 50, 742-754) are introduced to represent the substructures in rough structures and query molecules. In RDKit, ECFPs are also called Morgan fingerprints because the core idea of ECFPs comes from the Morgan algorithm (*J. Chem. Doc.*, 1965, 5, 107-113). The Tanimoto similarity is used to calculate the similarity between the ECFPs of each rough structure and its query molecule. The shared substructures between molecules can be measured by

molecular fingerprint similarity, and thus the quality of the rough structure can be evaluated. The mean molecular fingerprint similarity was calculated with the ECFPs of each molecule and its top 10 candidates for both FastEI and WCS. For 11,499 molecules in the test set, the violin plot in Fig. R9c show the mean molecular fingerprint similarities of FastEI and WCS, respectively. As can be seen from the distributions, the mean molecular fingerprint similarities of FastEI are larger than those of WCS. Similar to the previous paragraph, we also removed the shared rough structures. The mean molecular fingerprint similarities of the unique rough structures of FastEI and WCS are shown in Fig. R9d. It can be seen that the fingerprint similarities of the unique molecules from FastEI are significantly better than those of the unique molecules from WCS. This result indicates that the FastEI method is better at suggesting rough structures than the WCS method.

Fig. R9. Violin plots of mean matching values and molecular fingerprint similarities of FastEI and WCS. a the mean matching values between each molecule and its top 10 candidates of FastEI and WCS. **b** the mean matching values between each molecule and its unique candidates of FastEI and WCS. **c** the mean fingerprint similarities between each molecule and its top 10 rough structures of FastEI and WCS. **d** the mean fingerprint similarities between each molecule and its unique rough structures of FastEI and WCS.

Comment 2. Supplementary Fig. 3 is helpful to understand which data was used for model training/test. But multiple arrows confuse us whether they indicate data splitting. Also, it is difficult to see which data were converted to *in-silico* spectra and which step performed the conversion using NEIMS. Authors need to arrange the figure for the clarity.

Response: Firstly, we apologize for the confusion caused to the reviewers by the lack of clarity in Supplementary Fig. 3 of the manuscript. We agree with reviewer #3 that the multiple arrows complicate the data splitting, and Supplementary Fig. 3 should be arranged for clarity. By splitting the original Supplementary Fig. 3 into three subfigures, the problem of multiple arrows is avoided. It is also clarified which data have been converted to *in-silico* mass spectra by NEIMS. The three subfigures are detailed information on datasets used in the FastEI (Fig. R10a), construction of the *in-silico* libraries (Fig. R10b), and building of the Word2vec model (Fig. R10c). Fig. R10b clearly shows which data have been converted to *in-silico* mass spectra by the NEIMS model to build the *in-silico* library.

Fig. R10§§. Details and construction of datasets and libraries. **a** Detailed information on datasets used in the FastEI. **b** Construction of the *in-silico* libraries. **c** Building of the Word2vec model.

(a). Detailed information on datasets used in the FastEI. The details of the datasets are shown in Fig. R10a. There are three primary datasets in FastEI, which are NIST 2017, ChEMBL 28, and HMDB 5.0. We have explained them in detail below.

(a1). NIST 2017. The SMILES strings of the NIST 2017 dataset were downloaded from the NEIMS GitHub repository 33. It consists of 260,307 molecules. It was filtered using the five filtering rules (Supplementary Fig.3). The retained dataset is called f-NIST, which contains 255,821 molecules. They were split into the training set (232,826), validation set (11,496), and test set (11,499) strictly according to the rule of the NEIMS model by Wei et al. (*ACS Cent. Sci.*, 2019, 5, 700-708)

(a2). ChEMBL 28. The gzipped structure-data file (SDF) of ChEMBL 28 was downloaded from https://ftp.ebi.ac.uk/pub/databases/chembl/ChEMBLdb/releases/chembl_28/chembl_28.sdf.gz. The SMILES strings of molecules were read by RDKit (v2022.03.3). There are 2,066,376 molecules in the SDF file with mol files, of which 2,066,374 can be read using RDKit. After

§§ This is Supplementary Fig. 1 the revised Supporting information, and it is presented here for the convenience of reviewers.

filtering with five rules and deduplicating with f-NIST, the number of remaining molecules is 1,890,869. The resulting library is called *f*-ChEMBL.

(a3). **HMDB 5.0.** The zip archive of the SDF file of HMDB 5.0 was downloaded from its official website <https://hmdb.ca/system/downloads/current/structures.zip>. The SMILES strings of 217,759 molecules were read using RDKit (v2022.03.3). After filtering and deduplication with f-NIST and *f*-ChEMBL, the resulting library is called f-HMDB with 106,516 molecules.

(b). **Construction of the *in-silico* libraries.** All molecular ECFPs in the *f*-NIST, *f*-ChEMBL, *f*-HMDB, and the extra test set were put into the NEIMS model to obtain their predicted mass spectra. The NEIMS model was downloaded from https://storage.googleapis.com/deep-molecular-massspec/massspec_weights/massspec_weights.zip. The input of NEIMS is the ECFPs, and the RDKit package is used to calculate the ECFPs from SMILES strings. The output of the NEIMS model is a spectral vector representing the intensity at each *m/z*. As shown in Fig. R10b, the predicted spectra of molecules in *f*-NIST and *f*-ChEMBL were used to build the *in-silico* library, which included 2,146,690 molecules and their predicted spectra. The predicted spectra of molecules in *f*-HMDB and the extra test set were added to the *in-silico* library to build the expanded *in-silico* library, which included 2,253,216 molecules and their predicted spectra.

(c). **Word2vec model building and spectral embedding.** As shown in Supplementary Fig. 1c, based on gensim, a Python library, the Word2vec model was trained using 232,826 predicted spectra of the training set and 1,890,869 predicted spectra of the *f*-ChEMBL dataset. The measured spectra of 11,496 molecules in the validation set were used to validate the Word2vec model and optimize the hyperparameters of the Word2vec model. The measured spectra of 11,499 molecules in the test set were used to evaluate the performance of the Word2vec model.

Based on the clarification of Fig. R10, we completely rewrote the first three subsections of the *method* section: *Datasets; Prediction of spectra by NEIMS and construction of in-silico libraries; Word2vec model building and spectral embedding*. With the above revision, we believe that the data relationship is clear.

Comment 3. In addition to 2), were the test sets used for the evaluations converted to *in-silico* format by NEIMS? Or were they applied to the trained model as raw measured spectra without conversion? This difference is significant in the tool applicability, because the former case means less useful of this tool for data analysis of unknowns. This will be the main interest of the readers. Authors need to clarify this point.

Response: Sorry for the confusion due to the lack of data clarity yet. We fully agree with Reviewer #3. In practical application, only when the developed tool can complete the identification of unknown compounds without any prior information, and the usefulness of the developed tool can be demonstrated. **The raw measured spectra** are the only information that users need to enter. A candidate list can be obtained by simply entering the raw measured spectrum of the unknown into FastEI software. There is already **a large *in-silico* library with high molecular coverage**, which is included in the FastEI software. The conversion of unknowns to *in-silico* format by NEIMS is not required.

Comment 4. In addition to 2), supplementary Fig. 3 also seems to indicate that test data was used also for the training. In the main text, authors need to justify why the data used as test dataset are applicable to word2vec training and optimization, and expanded *in-silico* library. Otherwise, the model would be considered unvalidated with external data.

Response: Thanks for pointing out the misleading parts in supplementary Fig. 3. In this study, we strictly follow the "*Best practices in machine learning for chemistry*" (*Nat. Chem.*, 2021, 13, 505-508). The models used by FastEI have undergone rigorous testing with external data sets, which consist of unseen data to assess the performance of the models. The data sets of the NEIMS model and Word2vec model are described in detail below:

For the NEIMS model, we directly use the model trained by Wei et al. (*ACS Cent. Sci.*, 2019, 5, 700-708), which can be downloaded from this URL (https://storage.googleapis.com/deep-molecular-massspec/massspec_weights/massspec_weights.zip). Their SMILES strings and InChIKey can be downloaded from https://github.com/brain-research/deep-molecular-massspec/tree/main/training_splits. The NEIMS training set, the NEIMS validation set, and the

NEIMS test set were used to train the NEIMS model, optimize hyperparameters, and test its performance in Wei's paper, respectively. Due to the strict deduplication procedure, there are no common molecules among these three sets. The NEIMS test set is unseen by the NEIMS model before the testing. Therefore, it can be said that the NEIMS model is rigorously validated using the NEIMS test set. The training set (232,826), validation set (11,496), and test set (11,499) in Fig. R10b were obtained after filtering the NEIMS training set, NEIMS validation set, and the NEIMS test set with a five-step filtering procedure (Supplementary Fig. 1).

*“The SMILES strings of the NIST 2017 dataset were downloaded from the NEIMS GitHub repository³⁴. It consists of 260,307 molecules. It was filtered using the five filtering rules (Supplementary Fig. 3). The retained dataset is called *f*-NIST, which contains 255,821 molecules. They were split into the training set (232,826), validation set (11,496), and test set (11,499) strictly according to the rule of the NEIMS model by Wei et al⁹.”* (Line 30-Line 38, Page 9)

*“The NEIMS model was downloaded from its official repository²⁰. The input of NEIMS is the ECFPs, and the RDKit package is used to calculate the ECFPs from SMILES strings. The output of the NEIMS model is a spectral vector representing the intensity at each *m/z*.”* (Line 17-Line 22, Page 10)

For the Word2vec model constructing, three data sets are involved: the Word2vec training dataset, the validation set, and the test set. The Word2vec training dataset (Fig. R10c) consists of the training set (232,826 predicted spectra) and the *f*-ChEMBL dataset (1,890,869 predicted spectra). The validation set for the Word2vec model is the validation set in Fig. R10c (11,496 measured spectra collected from the NIST 2017), which is used to optimize the hyper-parameters of the Word2vec model. The test set for the Word2vec model is the test set in Fig. R10c (11,499 measured spectra collected from the NIST 2017), which is used to evaluate the performance of the Word2vec model. A rigorous deduplication procedure was also performed among these three sets. The test set is never used by the Word2vec model before testing. Therefore, the Word2vec model is strictly validated on the test set.

*“As shown in Supplementary Fig. 1c, based on gensim42, a Python library, the Word2vec model was trained using 232,826 predicted spectra of the training set and 1,890,869 predicted spectra of the *f*-ChEMBL dataset. The measured spectra of 11,496 molecules in the validation set were used to validate the Word2vec model and optimize the hyperparameters of the Word2vec model. The measured spectra of 11,499 molecules in the test set were used to evaluate the performance of the Word2vec model.”* (Line 37-Line 45, Page 10)

In addition to the models, there is an HNSW index for the *in-silico* library in FastEI. However, there is a fundamental difference between the HNSW indices and deep learning models. Deep learning models have learnable parameters that need to be trained with data. HNSW indices have no learnable parameters and do not need to be trained with data. Essentially, the HNSW index is an efficient data structure (the hierarchically multi-layer graph) to store and organize data, which is used to improve the speed of approximate nearest neighbor search (ANNS). In building the *in-silico* library, we only need to initialize an HNSW index and add the spectral embedding to the index. Therefore, there is no need to test the HNSW indices for overfitting problems using external data sets.

“HNSW indices have no learnable parameters and do not need to be trained with data. Essentially, the HNSW index is an efficient data structure (the hierarchically multi-layer graph) to store and organize data. It can be used to improve the speed of ANNS significantly.” (Line 79-Line 83, Page 10)

Finally, we explain why the test sets should be added to the *in-silico* library and the expanded *in-silico* library. In FastEI software, there are two files for an *in-silico* library: an HNSW index and an SQLite file. The HNSW index takes full advantage of the hierarchically multi-layer graph architecture to store and organize Word2vec embeddings of *in-silico* mass spectra for efficient ANNS. The SQLite file store the identifier, SMILES string, and *in-silico* mass spectrum predicted by NEIMS of each compound. Both the HNSW index and the SQLite file are based on **the *in-silico* mass spectra predicted from the molecular structures by NEIMS**. It can be seen that FastEI just stores the molecular structure information of compounds in the HNSW index file and the SQLite

file. The molecular structure information of the test set must be added to the *in-silico* library. Otherwise, the identification of molecules in the test sets cannot be achieved due to coverage issues. It is also important to emphasize that the experimental mass spectra of the test sets are not used in the *in-silico* library. The measured mass spectra, analyzing the standard compounds with the GC-MS instrument, are used as inputs to the FastEI software. The input experimental mass spectrum is transformed into the Word2vec embedding. The HNSW index can be used to efficiently find the *in-silico* mass spectral embeddings (their corresponding molecular structures) close to this experimental mass spectral embedding. Essentially, the FastEI software compares the experimental mass spectra of the compounds acquired by the GC-MS instrument with the molecular structure information of the compounds in the *in-silico* library.

In conclusion, only the molecular structure information has been used when adding the test sets into the *in-silico* library. The experimental mass spectra of the test sets are utilized as inputs of FastEI when performing compound identification. It can be considered that such a setting is reasonable in compound identification.

Comment 5. Authors require to discuss why the Word2vec improve the accuracy compared with WCS in the section of discussion.

Response: We strongly agree with this comment from reviewer #3. If FastEI is better than WCS, the reasons behind the phenomenon are more important than the results. It allows users to have more confidence in FastEI and also gives developers direction to design better methods. In light of this initial intention, we conducted an in-depth investigation of the results of FastEI and WCS methods on the test set. The specific procedure of the investigation is as follows:

(a). **Recall@1 molecules.** The test set and the *in-silico* library were selected as the inputs to benchmark FastEI and WCS methods. After the spectral matching, the recall@1 of FastEI and WCS methods can be obtained. As can be seen from Table R4, the recall@1 for FastEI and WCS are 36.7% and 30.2%, respectively. The difference in recall@1 between these two methods is 6.5%, and there must be some reason for such a significant difference.

Table R4^{***}. The *mainlib* test set spectrum matching performance on the *in-silico* library.

Method	recall@1 (%)	recall@10 (%)	Run time per spectrum matching (s)
FastEI	36.7	80.4	0.0042
WCS	30.2	73.5	2.4849

(b). **Venn diagram.** The 11,499 experimental spectra (collected from the NIST main library) of molecules in the test set matched the *in-silico* library using the FastEI and WCS methods. As shown in Fig. R11, the target molecules of the 2,812 spectra rank top 1 in both FastEI and WCS. The target molecules of the 1,413 spectra rank top 1 only by FastEI, and the target molecules of the 659 spectra rank top 1 only by WCS. We explore the accuracy improved by Word2vec-based spectral embeddings compared with the spectral binning vectors by taking a deeper look at these 1,413 spectra and their library matching results of FastEI.

^{***} This is part of Table 1 the revised manuscript, and it is presented here for the convenience of reviewers.

Fig. R11^{†††} Venn diagram of the recall@1 molecules of FastEI and WCS.

Fig. R12^{†††} Left: box plot of the spectral embedding similarities of top1 candidates is compared with the embedding similarities of other candidates (the top10 except for top1); Right: boxplot of the spectral binning similarities of top1 candidates are compared with the spectral binning similarities of other candidates (the top10 except for top1).

(c). **Similarities of unique recall@1 molecules of FastEI.** To investigate the difference matching performance between the spectral binning vectors and Word2vec embeddings, the predicted spectra of the top 10 candidates of these 1,413 experimental spectra using FastEI are retrieved from the *in-silico* library. All the predicted and experimental spectra can be transformed into spectral embeddings by the Word2vec model. For each experimental spectra, the spectral binning similarity between the experimental spectrum and its candidate predicted spectrum has been calculated by the cosine similarity method. The embedding similarity between experimental embedding and its candidate predicted spectrum embedding has also been calculated by the cosine similarity method. The boxplot in Fig. R12 shows the spectral embedding similarities of top1 candidates are compared with the embedding similarities of other candidates (the top10 except for top1); the spectral binning similarities of top1 candidates are compared with the spectral binning similarities of other candidates (the top10 except for top1). The result shows that the embedding similarity has **better distinguishability** than spectral binning similarity. It can be said that the embedding similarity can make the same molecules have high cosine similarity and different molecules have low cosine similarity.

^{†††} This is Supplementary Fig. 5 in the revised manuscript, and it is presented here for the convenience of reviewers

^{†††} This is Supplementary Fig. 6 in the revised manuscript, and it is presented here for the convenience of reviewers

Fig. R13^{§§§} UMAP plots for all the unique recall@1 molecules of FastEI. **a** the UMAP plot of 1,413 pairs of predicted and experimental spectral binning vectors. **b** the UMAP plot of 1,413 pairs of predicted and experimental spectral embeddings. **c** the UMAP plot of 15 pairs of predicted and experimental spectral binning vectors randomly selected from 1,413 experimental and predicted spectral pairs. **d** the UMAP plot of these randomly selected 15 pairs of predicted and experimental spectral embeddings.

(d). UMAP plots of unique recall@1 molecules. These 1,413 experimental spectra and their predicted spectra (collected from the *in-silico* library) are represented by binning vectors and Word2vec-based embeddings. The UMAP method is used to reduce the dimensionalities for visualization. The UMAP plot of these predicted and experimental spectral binning vectors is shown in Fig. R13a, and the UMAP plot of these predicted and experimental spectral embeddings is shown in Fig. R13b. By comparing Fig. R13a with Fig. R13b, the predicted and experimental spectral embeddings have a more uniform spatial distribution and less aggregation than spectral binnings. The UMAP plot of 15 pairs of predicted and experimental spectral binning vectors randomly selected from 1,413 experimental and predicted spectrum pairs is shown in Fig. R13c. The UMAP plot of these 15 pairs of the predicted and experimental spectral embeddings is shown in Fig. R13d. By comparing Fig. R13c with Fig. R13d, the relative distance between predicted and experimental spectral embeddings of the same molecule is significantly smaller than the distance between predicted and experimental spectral binning vectors. The result shows that the deviation between the predicted mass spectra and the experimental mass spectra is significantly reduced after the Word2vec-based spectral embeddings.

(e). A possible explanation. The *in-silico* mass spectra are predicted from the SMILES strings of the molecules, and the experimental mass spectra are acquired on the GC-MS instruments. Due to the different inputs (SMILES strings v.s. standard compounds) and generation methods (NEIMS model v.s. mass spectrometry), there are some deviations between the *in-silico* mass spectra and the experimental mass spectra. A Word2vec model was adapted to learn meaningful representations of mass spectra and get d-dimensional embeddings for library matching. The deviation between the *in-silico* embeddings and the experimental embeddings can be significantly reduced after the

^{§§§} This is Supplementary Fig. 7 in the revised manuscript, and it is presented here for the convenience of reviewers

Word2vec processing. The reduction in deviation significantly enhances the distinguishability between the *in-silico* and experimental embeddings, thus improving the accuracy of compound identification. After the above analysis, we have derived a possible explanation. The related content has been added to the discussion section as follows:

*“The possible explanation for the matching accuracy improved by Word2vec-based spectral embeddings compared with the spectral binning vectors is that the Word2vec model reduces the dimensions of the spectra and embeds them into chemically meaningful representations. As shown in Supplementary Note 2, Fig. 5, and Fig. 6, the embedding similarity can make the same molecules have high cosine similarity, and different molecules have low cosine similarity. Arguably, embedding similarity correlates better with structural similarity than binning similarity. Meanwhile, the *in-silico* mass spectra are predicted from the molecular structures, and the query mass spectra are acquired on the GC-MS instruments. There exist deviations between the *in-silico* mass spectra and the experimental mass spectra. As shown in Supplementary Note 2, Fig. 5, and Fig. 7, this deviation can be significantly reduced after the Word2vec coding the predicted and the measured spectra. Therefore, the Word2vec spectral embedding improves the accuracy of compound identification.”*
(Line 70-Line 92, Page 8)

Comment 6. The authors should mention the reason why the expanded library deteriorate the matching accuracy in the section of discussion. Can this be overcome by reconstructing the models based on the new dataset including the additional dataset?

Response: This comment is very insightful. It can provide a solid foundation for expanding the *in-silico* library. In this response, we first explore the reason for the decrease in matching accuracy when expanding the *in-silico* library. Then, we retrain the Word2vec model using a new training set to see whether or not the decrease in the performance of library expansion can be circumvented.

(a). Accuracy decreasing on the expanded library. To clarify the reasons for the accuracy decrease, it is necessary to find out those molecules whose accuracies decrease on the expanded library. The mass spectra of the molecules in the test set were embedded by the Word2vec model and then matched with the *in-silico* library and the expanded library using FastEI, respectively. After the matching, we obtained the recall@1 molecules of the *in-silico* library and the recall@1 molecules of the expanded library, respectively. From the Venn diagram in Fig. R14a, it can be seen that the two libraries have 4183 common recall@1 molecules, and there are 42 unique recall@1 molecules for the *in-silico* library and 0 unique recall@1 molecules for the expanded library, respectively. The calculation shows that the percentage of common recall@1 molecules is 99%, indicating that the expansion of the *in-silico* library has little effect on the recall@1 metric. For each molecule in the 42 unique recall@1 molecules, when matching the *in-silico* library, its rank decreases when matching the expanded library. Their rankings are shown in Fig. R14b, where it can be seen that most of the molecules (83%) drop only 1-2 positions. Two compounds were not found in the top 100 in the extended library because of the introduction of lipid molecules from the HMDB library. The mass spectra of these three molecules share many peaks with the lipid molecules. So a large number of lipid molecules were matched.

For each molecule in the 42 unique recall@1 molecules when matching the *in-silico* library, there is a false positive recall@1 molecule when matching the expanded library. For the convenience of presentation, the former molecule is referred to as the true positive molecule (TP molecule) and the latter molecule as the false positive molecule (FP molecule). The corresponding TP molecule and FP molecule make up a molecular pair. For the TP and FP molecules, their molecular structures, experimental spectra, *in-silico* spectra, and spectral embeddings were retrieved from the expanded library and the test set. The molecular structure similarity, spectral similarity, and spectral embedding similarity were calculated for each molecular pair. As can be seen in Fig. 14c, there are molecular pairs with high structural similarities or spectral similarities, which leads to high embedding similarities. Therefore, the reason for the decrease in accuracy in the expanded library is the introduction of molecules with very similar structures or mass spectra during the expansion process.

Fig. R14 Analysis of accuracy decrease when matching the expanded library. **a** Venn diagram of the recall@1 molecules when matching the *in-silico* library and the expanded library. **b** Bar plot of the rank difference of the unique recall@1 molecules between the *in-silico* library and the expanded library. **c** Boxplot of the molecular structure, experimental spectral, *in-silico* spectral, and spectral embedding similarities for the unique recall@1 molecules.

Fig. R15 Venn diagram of the recall@1 molecules of the Word2vec model trained on the Word2vec training set and the new dataset.

(b). Retraining models with the new dataset. Based on the suggestion of reviewer #3, we constructed a new dataset by adding predicted spectra of additional molecules to the Word2vec training set. Then, a new Word2vec model was trained using this dataset. The model was used to convert the spectra into their spectral embeddings. The new HNSW indices were built for both the *in-silico* library and the expanded library. The recall@1 molecules were obtained by analyzing the test set using the new Word2vec model and HNSW indices. From the Venn diagram in Fig. R15, it can be seen that the two libraries have 3881 common recall@1 molecules (93%). There are 302 unique recall@1 molecules for the expanded library based on the original Word2vec model and 290 unique recall@1 molecules for the expanded library based on the retrained Word2vec model, respectively. These results indicate that the retrained Word2vec model has slightly decreased the recall@1 metric. The possible explanation is that the Word2vec training set is large and diverse enough to train a model with good predictive ability and achieve comparable results on the test sets. Furthermore, it takes about 48 hours to train the Word2vec model. In addition, all mass spectra in the *in-silico* library need to be embedded using the model, the HNSW index should be recreated, and the spectral embeddings should be added to the index. These two steps take about 50 hours. One can see that it is time-consuming to retrain the model using the new dataset, and the accuracy is not improved. Therefore, the Word2vec model is rarely retrained with the new dataset when expanding the library. In most cases, the library expansion can be simply done with the existing NEIMS model and Word2vec model for mass spectral prediction and spectral embedding, respectively. Then the spectral embeddings are added to the HNSW index.

Specific Comments

Comment 7. Resolution of Figure 1 should be improved.

Response: Thanks for the comment on the figure quality. We have improved the resolution of all the figures according to the suggestion from reviewer #3. All the figures were exported in lossless TIFF format at 300 by 300 resolution. To avoid compression of inserted figures by Word, we have turned on the option "do not compress images in file". After the above operations, the resolution of all the figures in the article has been significantly improved, and the clarity was significantly increased. During the revision process, we also found that the fonts in some figures were too small, which significantly affected the quality of the figures. Therefore, we also enlarge the font in the figures for better readability. The revised Fig. 1 is provided here as Fig. R16 for the convenience of the reviewers.

Fig. R16** Overview of FastEI.** a Flowchart of the FastEI and WCS method for spectrum matching against the *in-silico* spectral library.

Comment 8. ClassyFire should be mentioned in the first mention.

Response: Thanks to reviewer #3 for the reminder on the correct position of in-text citations. We have added the in-text citation to ClassyFire when it appears the first time. Meanwhile, we have also checked the in-text citation of other methods.

“Meanwhile, Fig. 2a shows the classes of molecules in the test set predicted by ClassyFire²¹” (Line 12-Line 13, Page 3)

Comment 9. The order and place of Fig 2e and 2d should be exchanged, because Fig.2d is mentioned earlier in main text.

Response: Thanks for your suggestion. Fig. 2e is a subfigure that occupies two columns and is not well adjusted to the position of Fig. 2d. Therefore, we have changed the section order of “Contribution of Word2vec embeddings and HNSW” and “Applicability of FastEI to spectra from different sources” in the manuscript. In this way, they are mentioned in the manuscript in the correct order.

**** This is Fig.1 in the revised manuscript, and it is presented here for the convenience of reviewers.

Comment 10. Authors should describe how to prepare query spectrum in csv format for users. Also, it is more user friendly to include example csv data into FastEI download folder.

Response: This is an excellent suggestion. The input format of the mass spectra needs to be precisely defined, so the users can prepare the mass spectra for the compound to be identified. We have taken the following steps, which allow users to convert their mass spectra into CSV format easily.

1	29	1.9
2	30	0.38
3	31	0.47
4	32	0.31
5	33	0.04
6	34	0.03
7	36	0.13
8	37	1.9
9	38	4.51
10	39	4.01
11	40	0.79
12	41	1.47
13	42	0.43
14	43	1.39
15	44	0.08

Fig. R17 The definition of CSV format for storing mass spectra in FastEI.

(a). **CSV format definition.** The CSV is an abbreviation for comma-separated values. Each CSV file has multiple rows, each row with multiple values, which are separated by commas. Specifically for storing mass spectra in CSV format, each mass spectrum has multiple peaks, and each peak has its mass-to-charge ratio and intensity. Therefore, we save each peak of a mass spectrum as a line in the CSV file. The first value in a line is the mass-to-charge ratio of a peak, and the second value is the intensity of this peak. It should be emphasized that the mass-to-charge ratio must be an integer, and each intensity must be normalized by its max intensity. The definition of CSV format for storing mass spectra in FastEI is shown in Fig. R17. We have also uploaded the CSV format definition to the Github repository of FastEI in markdown format, and the link is https://github.com/Qiong-Yang/FastEI/blob/main/ms_to_csv/README.md.

(b). **Common formats.** All mass spectra can be converted to the defined CSV format as long as they have m/z and intensity values. In FastEI, it supports the commonly used formats such as the MSP format, JCAMP-DX (JDX) format, and TXT format of Shimadzu GCMSsolution. The MSP format is the format for mass spectra in NIST MS Search software. JCAMP-DX are text-based file formats created by Joint Committee on Atomic and Molecular Physical Data (JCAMP) for storing and exchanging spectroscopic data. The TXT format is the file format for exporting mass spectra from Shimadzu GCMSsolution.

(c). **Jupyter notebook.** In the FastEI Github repository, we provide functions to convert MSP, JDX, and Shimadzu GCMSsolution TXT formats to CSV format. To demonstrate the usage of these functions, we provide an example in the form of a Jupyter Notebook. It is available at https://github.com/Qiong-Yang/FastEI/blob/main/ms_to_csv/example.ipynb.

(d). **Converter.** A conversion script has also been implemented for user-friendliness. Given a directory name, this script can convert all supported mass spectrum files in the directory to CSV format. It is available at https://github.com/Qiong-Yang/FastEI/blob/main/ms_to_csv/converter.py. Furthermore, the improved software supports the input of different formats (MSP format, JDX format, and CSV format) of the experimental spectrum for library matching.

(e). **Example CSV files.** Some example CSV files have been included in both the GitHub repository of FastEI and the download folder of FastEI. The mass spectra of ten compounds in the extra test set were acquired by a GCMS-QP2010 Ultra instrument and exported in the Shimadzu GCMSsolution TXT format. These Shimadzu GCMSsolution TXT files are available at https://github.com/Qiong-Yang/FastEI/tree/main/data/extra_test_set with the “.txt” suffix. They were converted to CSV format using the conversion script. The converted CSV files are also available at https://github.com/Qiong-Yang/FastEI/tree/main/data/extra_test_set with the “.csv” suffix. When generating the installer of FastEI with InnoSetup (v6.2.1), the converted CSV files are included in the installer. The FastEI installer is uploaded into the download folder of FastEI at <https://github.com/Qiong-Yang/FastEI/releases>. After downloading and installing FastEI, users can find these CSV files in the *data* subdirectory of its root installation directory.

Comment 11. Figure 4 should include caption letter (a,b,c,d) inside of the figure.

Response: We agree with reviewer #3 that there should be caption letters in Fig. 4 for referencing the different functional parts of the screenshot of FastEI in the manuscript. Therefore, four caption letters (a, b, c, and d) have been added to Fig. 4 for the data loading, query spectrum loading, matching candidates display, and progress bar parts, respectively. The revised Fig. 4 is provided here as Fig. R18 for the convenience of reviewers.

Fig. R18^{†††} Screenshot of FastEI. **a** loading database, index, and model. **b** loading query spectra to be identified. **c** function buttons for matching the *in-silico* library, viewing the candidates, and plotting the mass spectra and molecular structures. **d** a progress bar for indicating the time-consuming operations.

Furthermore, the spectrum matching and results displayed have been optimized for ease of use. We have improved FastEI by removing the unnecessary *confirm*, *view*, and *plot* buttons in the query spectrum loading part. In the query spectrum loading part (Fig. R18b), users can load the CSV files using the query button, and each loaded query spectrum can be clicked directly to perform the spectrum matching. After the spectrum matching, the matching results will be shown in the matching candidate display part (Fig. R18c). When clicking on a candidate, the query spectrum of

^{†††} This is Fig.4 in the revised manuscript, and it is presented here for the convenience of reviewers.

the unknown and the *in-silico* spectrum of the candidate will show in the spectrum tab for comparison. The molecular structure of the candidate can be shown by clicking on the structure tab. As the GUI of FastEI has been updated, the demo video has also been updated in the GitHub repository of FastEI.

Comment 12. I understand that the duplication was removed within the database, but what if there are duplication among the database of NIST, ChEMBL, NEIMS-training set, and HMDB?

Response: It is necessary to emphasize that all the datasets (libraries) are strictly deduplicated within each dataset (library) and among all the datasets (libraries) to ensure the uniqueness of the molecules in the library for FastEI, so there are no above problems in our study. It may be that Supplementary Fig. 3 in the original manuscript is not well drawn. In this revision, we have redrawn Supplementary Fig. 3 to make the relationship among the datasets more clear. There are four data sources, which are NIST 2017, ChEMBL 28, HMDB 5.0, and the extra test set, respectively. The details of each dataset were provided in the *Methods* section and in response to comment 2.

REVIEWER COMMENTS

Reviewer #1 (Remarks to the Author):

I am mostly satisfied with the authors' responses to the reviewer comments, and thank the authors for their extensive follow ups.

I have a few more minor comments to help prepare the manuscript further for submission.

- In figure 2, I find the text of the axis titles very small and difficult to read. Please increase the font size of these figures.

- Regarding the structural groupings with ClassyFire, thank you for your analysis of performance based on compound types according to the embeddings. I'm not quite sure if I follow the analysis in Figure R13c and d addressing reviewer 3's comment. The caption and the text indicate that the UMAP contains a comparison of experimental and predicted spectra for 15 pairs of binning vectors, but it appears that there are many more than 15 points in the UMAP representation?

- In the main text, regarding the slight decrease in recall performance upon library expansion (pg. 8 lines 49-50), I found the use of the term 'richer' confusing. It might be more clear to state that 'the expanded library has more compound structures than the original library. You might even compare the number of compounds in of chemical classes between the original training library for NEIMS and the new expanded library to help support this point.

- If possible, it would be interesting to view a UMAP or another representation showing spectral embeddings according to the Word2Vec model colored by structural class (perhaps given by ClassyFire).

- Regarding the use of EI-MS in metabolites: Thank you for providing the references indicating the use of GC/MS in identifying metabolites. I believe that GC/MS and in particular EI-MS is used in the identification of some metabolites, but ESI is the more common ionization method. For example, observing the MassBank of North America database (<https://mona.fiehnlab.ucdavis.edu/spectra/statistics>), I find only 67 spectra out of 695,424 spectra are EI-MS spectra (<https://mona.fiehnlab.ucdavis.edu/spectra/browse?query=metaData%3Dq%3D%27name%3D%3D%22ionization%22%20and%20value%3D%3D%22EI%22%27&text=&size=10>).

That said, I agree with the authors that Fast-El is still useful for metabolites when using GC/MS spectroscopy, and its usefulness could be expanded to other mass spectroscopy ionization strategies.

Reviewer #3 (Remarks to the Author):

Please see attached.

Responses to reviewer comments

Reviewer #1:

I am mostly satisfied with the authors' responses to the reviewer comments, and thank the authors for their extensive follow ups.

I have a few more minor comments to help prepare the manuscript further for submission.

Comment 1. In figure 2, I find the text of the axis titles very small and difficult to read. Please increase the font size of these figures.

Response: Thanks for the comment on the font size of Fig. 2. We have increased the font size from 24 to 36, and the readability of the text in Fig. 2 is significantly improved. The revised Fig. 2 is provided here as Fig. R1 for the convenience of the reviewers.

Fig R1 *. Information of datasets and performance of FastEI. **a** The molecule classes predicted by ClassyFire for the test set. **b** The visualization of the ECFPs of 240,000 molecules randomly selected from *f*-ChEMBL and 232,826 molecules from the training set by UMAP. **c** The spectrum matching time of FastEI and WCS on libraries with different sizes. **d** The contribution of Word2vec embeddings and HNSW to FastEI (WCS: weighted binning + cosine similarity, EC: embeddings + cosine similarity, BH: weighted binning + HNSW, FastEI: embeddings + HNSW). **e** The performance of FastEI and WCS on the test set in terms of recall rates at different top x levels.

* This is Fig. 2 of the revised manuscript, and it is presented here for the convenience of reviewers.

Comment 2. Regarding the structural groupings with ClassyFire, thank you for your analysis of performance based on compound types according to the embeddings. I'm not quite sure if I follow the analysis in Figure R13c and d addressing reviewer 3's comment. The caption and the text indicate that the UMAP contains a comparison of experimental and predicted spectra for 15 pairs of binning vectors, but it appears that there are many more than 15 points in the UMAP representation?

Response: Thanks for this professional comment from Reviewer #1. We agree that a statistical comparison of all samples is more convincing than highlighting some representative samples due to its comprehensiveness and objectivity. Therefore, the relative Euclidean distances (RED) between experimental and predicted spectral pairs have been calculated for all the 1,413 unique recall@1 molecules of FastEI using the UMAP dimensionality-reduced data of both the binning vectors and the embeddings. The formula of RED is shown below:

$$RED = \frac{\sqrt{\left(U_{exp_i}^1 - U_{pred_i}^1\right)^2 + \left(U_{exp_i}^2 - U_{pred_i}^2\right)^2}}{\text{Max}\left(\sqrt{\left(U_j^1 - U_k^1\right)^2 + \left(U_j^2 - U_k^2\right)^2}\right)} \quad (\text{R-1})$$

Here, $U_{exp_i}^1$ and $U_{pred_i}^1$ are the values of experimental and predicted spectra in the first-dimensional coordinate after dimensionality reduction by UMAP. $U_{exp_i}^2$ and $U_{pred_i}^2$ are the values of experimental and predicted spectra in the second-dimensional coordinate after dimensionality reduction by UMAP. The subscripts j and k represent different molecules. The RED is the Euclidean distance between the experimental and predicted spectra of a given molecule divided by the distance between the two farthest points in the UMAP reduced dimensional space. This farthest distance is calculated between different molecules without distinguishing the experimental spectrum and the predicted spectra. We provide a Jupyter Notebook for plotting Fig. R2, which is available at:

https://github.com/Qiong-Yang/FastEI/blob/main/additional%20instructions/Research_on_spectral_word2vec_embedding_mechanism.ipynb.

As shown in Fig. R2c, all the RED of the binning vectors and the embeddings are visualized in their respective violin plots. It shows that the spectral embedding can narrow the RED between the predicted and experimental pair compared to the binning vector. The standard deviation of RED of embedding (0.069) is significantly lower than that of RED of binnings (0.126). As stated by Reviewer #3, the Word2vec model removes noisy bins that can interfere the possible discrimination (i.e., noisy bins weaken power of the discrimination). It is clear that Fig. R2c is more convincing than the original Supplementary Fig. 7c and 7d. Hence, we have replaced the original Supplementary Fig. 7c and 7d with Fig. R2c in this revision to present the comparison results in a more comprehensive and objective manner.

Fig. R2.[†] Plots for the distance between the experimental spectrum and their predicted spectrum of all the unique recall@1 molecules of FastEL. **a** the UMAP plot of 1,413 pairs of predicted and experimental spectral binning vectors. **b** the UMAP plot of 1,413 pairs of predicted and experimental spectral embeddings. **c** the violin plot of relative Euclidean distance (RED) between predicted and experimental spectral Word2vec embeddings and binning vectors of all 1413 molecules. M-RED represents the median of RED, and the SD-RED represents the standard deviation of RED.

Comment 3. In the main text, regarding the slight decrease in recall performance upon library expansion (pg. 8 lines 49-50), I found the use of the term 'richer' confusing. it might be more clear to state that 'the expanded library has more compound structures than the original library. You might even compare the number of compounds of chemical classes between the original training library for NEIMS and the new expanded library to help support this point.

Response: We completely agree with Reviewer #1 that “more compound structures” is more precise than “richer”. As stated by the reviewer, the chemical classes of the original *in-silico* library and the expanded library can be compared to support the “more compound structures” point. We counted the compound species of the extended library and the *in-silico* library by ClassyFire separately. The result is shown as Venn diagrams in Fig. R3. It can be found that 7

[†] This is Supplementary Figure 6 in the revised manuscript, and it is presented here for the convenience of reviewers.

new classes of compounds appear in the extended library. More compound structures here mean that more isomers or structurally similar candidates are obtained for the same molecular when matching the expanded library. Consequently, the correct structures of some queries are ranked lower in the list of spectral matching. The sentence has been revised, and it is excerpted here for the convenience of reviewers:

“The slight decrease in the matching accuracy should be attributed to the expanded library having more compound structures than the *in-silico* library. As shown in Supplementary Figure 3, 7 new classes (ClassyFire) of compounds appear in the expanded library relative to the *in-silico* library.” (Line 78-Line 84, Page 5)

Fig. R3.[‡] Venn diagram of the compound classes in the *in-silico* library and the expanded library. Seven new classes in the expanded library: Homogeneous metalloid compounds, Dioxoles, Oxirenes, Molybdopterin dinucleotides, Sulfines, Glycinamide ribonucleotides, Organosulfenic acids and derivatives.

Meanwhile, the chemical classes of the NEIMS training set and the new compounds added to the *in-silico* library were also compared. This comparison is primarily used to support that compounds used for library expansion can be predicted using the NEIMS model. As shown in Fig. R4, the number of chemical classes of compounds in the NEIMS training set is 427, which indicates that the NEIMS training set covers a wider range. The number of chemical classes of the newly added molecules is 333. There are 311 compound classes presented in both the NEIMS training set and the added compounds. There are 93% of the compound classes in the added molecules included in the compound classes of the NEIMS training set. Therefore, it is reasonable to use the NEIMS model to predict the EI-MS spectra of these added molecules.

Fig. R4. Venn diagram of the compound classes in the NEIMS training set and the expanded molecules added to the *in-silico* library.

[‡] This is Supplementary Figure 3 in the revised manuscript, and it is presented here for the convenience of reviewers.

Comment 4. If possible, it would be interesting to view a UMAP or another representation showing spectral embeddings according to the Word2Vec model colored by structural class (perhaps given by ClassyFire).

Response: This is an excellent and constructive suggestion. We have visualized the Word2vec embeddings and the spectral binning vectors of the experimental spectra in the test set. Their UMAP plots are shown in Fig. R5a and b, respectively. We provide a Jupyter Notebook for plotting Fig. R5, and it is available at:

https://github.com/Qiong-Yang/FastEI/blob/main/additional%20instructions/Colored_by_superclasses.ipynb.

Meanwhile, ClassyFire has been used to classify molecules in the test set into 21 chemical superclasses and 254 chemical classes. With so many chemical classes, it will complicate the figure. Therefore, molecules in the UMAP plots were colored by their chemical superclasses. As shown in Fig. R5a, there is a tendency for molecules to aggregate by their chemical superclasses in the UMAP space after Word2vec embedding. From Fig. R5b, it can be seen that the molecules are randomly distributed in UMAP space and are unrelated to their chemical superclasses. It can be concluded that the Word2vec embeddings are more relevant to the chemical superclasses than the spectral binning vectors by comparing Fig. R5a and b.

“By comparing Supplementary Figure 8a and b, the Word2vec embeddings are more relevant to the chemical superclasses than the spectral binning vectors.” (Line 12-Line 14, Page 10)

Fig. R5[§]. UMAP plots of different representations of the experimental spectra in the test set. **a** the UMAP plot of Word2vec embeddings colored by chemical superclasses. **b** the UMAP plot of spectral binning vectors colored by chemical superclasses.

[§] This is Supplementary Figure 8 in the revised manuscript, and it is presented here for the convenience of reviewers.

Comment 5. Regarding the use of EI-MS in metabolites: Thank you for providing the references indicating the use of GC/MS in identifying metabolites. I believe that GC/MS and in particular EI-MS is used in the identification of some metabolites, but ESI is the more common ionization method. For example, observing the MassBank of North America database (<https://mona.fiehnlab.ucdavis.edu/spectra/statistics>), I find only 67 spectra out of 695,424 spectra are EI-MS spectra (<https://mona.fiehnlab.ucdavis.edu/spectra/browse?query=metaData%3Dq%3D%27name%3D%3D%22ionization%22%20and%20value%3D%3D%22EI%22%27&text=&size=10>). That said, I agree with the authors that Fast-EI is still useful for metabolites when using GC/MS spectroscopy, and its usefulness could be expanded to other mass spectroscopy ionization strategies.

Response: We appreciate the reviewer's approval of our work. It is true that there are much more ESI-MS spectra than EI-MS spectra in MoNA. Meanwhile, we have to notice one thing: **to contrast the number of compounds is more meaningful than to contrast the number of mass spectra in database.**

The 205,044 experimental mass spectra** in MoNA were used as an example for the subsequent comparison. Among them, there are 13,572 EI-MS spectra and 136,055 ESI-MS/MS spectra. Those 13,572 EI-MS spectra are derived from **8,814 compounds**. The 136,055 ESI-MS/MS spectra are derived from **10,549 compounds**. The main reason for such a difference between the number of ESI-MS/MS mass spectra and the number of compounds is that: (a) higher reproducibility and stability of EI-MS spectra than ESI-MS; (b) the ESI-MS experiment is more complicated than EI-MS. When a compound is analyzed by ESI-MS, multiple MS/MS mass spectra can be generated due to factors such as ionization modes (positive mode and negative mode), collision energy (10V, 20V, and 40V), adduct types ($[M+H]^+$, $[M+Na]^+$, $[M+NH_4]^+$, $[M-H]^-$, $[M+HAc-H]^-$, $[M+CH_3COO]^-$, $[M+HCOO]^-$, $[M+FA-H]^-$), fragmentation methods (CID and HCD), and mass spectrometers (QqQ, QTOF, IT-TOF, and Orbitrap). For example, Catechin has 101 ESI-MS/MS spectra in MoNA. After statistically analyzing the downloaded ESI-MS/MS spectra, there is an average of 13 ESI-MS/MS spectra for a compound in MoNA, and there is only an average of 2 EI-MS spectra for a compound. A Jupyter notebook has been created with procedures for counting the number of compounds in the MoNA:

https://github.com/Qiong-Yang/FastEI/blob/main/additional%20instructions/Count_the_number_of_compounds_in_MoNA.ipynb

So, the difference of the number of compounds is insignificant between EI-MS and ESI-MS/MS in the MoNA. This shows that EI-MS is one of the widely used techniques in metabolomics and other fields, such as environment and food science.

** It is a commonly referenced predefined query of the MoNa database, and its URL is:
[https://mona.fiehnlab.ucdavis.edu/spectra/browse?query=not\(exists\(tags.text%20in%20\(%27In-Silico%27\)\)\)](https://mona.fiehnlab.ucdavis.edu/spectra/browse?query=not(exists(tags.text%20in%20(%27In-Silico%27))))

Reviewer #3:

General Comments:

Authors made huge efforts for the revision. They have successfully addressed almost all of the unclear points that arose in my previous review. Also, the revised contains interesting new findings. Just a few points to be clarified have newly arisen in the revised manuscript, especially in added/modified supplemental materials. Please see below

Specific Comments:

Comment 1. P8 L70-92 “The possible explanation...compound identification.” The supplementary Note2, Fig. 5-7 showed us interesting findings. In my opinion, the data analyses through the supplementary tell us that the Word2vec model successes to mine important features that actually efficient for structure “discrimination”. At the same time the model removes noisy bins that can interfere the possible discrimination (i.e., noisy bins weaken power of the discrimination). From another aspect, the conversion of spectral bins to embeddings by Word2vec successfully re-scale the chemical space so that relative distances among the chemicals are expanded. These result in improved accuracy of true/false candidate selection, as shown in supplementary Fig. 6. I advise authors to adjust the sentence so that the readers can precisely and directly catch the findings regarding why the Word2vec model works well for the accurate matching.

Response: We really appreciate this **INSIGHTFUL** explanation from Reviewer #3. We have adjusted our explanation accordingly. The revised version is more precise and clearer. It is excerpted here for the convenience of reviewers:

“As shown in Supplementary Figure 4 to Figure 6 and Supplementary Note 2, the Word2vec model successfully mines important features that are efficient for structure “discrimination” and improves the accuracy of true/false candidate selection. There are 1,413 target molecules in the test set ranked top1 by FastEI, and ranked outside of the top1 by WCS. After Word2vec spectral embeddings, the same molecules have high cosine similarity, and different molecules have low cosine similarity. The relative distance is narrowed between the predicted and experimental spectra compared to the binning vectors. The possible explanation for these performance improvements is that the conversion of spectral bins to embeddings by Word2vec successfully re-scales the chemical space, and relative distances among the chemicals are expanded. At the same time, the model removes noisy bins that may interfere with the possible discrimination (i.e., noisy bins weaken the power of the discrimination.” (Line 64-Line 85, Page 8)

Comment 2. Supplementary Fig. 7 (a, b): It is difficult to find the plot aggregation that authors suggest from Fig. 7. I recommend to calculate the area of convex hull of the plots (or density value, such as plot numbers in unit area on the convex hull), and provide the value differences between Fig. 7a and 7b.

And Comment 3. Supplementary Fig. 7 (c, d): Similar with above, the relative distance between predicted and experimental spectral binning vectors should be calculated and provided (e.g., as median \pm SD).

Response: A similar question was also mentioned in Comment 2 of Reviewer #1. Indeed, in the original Supplementary Fig. 7 (a, b), a direct comparison of their aggregation is not a good way to show the advantages of the FastEI method. A good representation should make the distance between the predicted and experimental mass spectra of the same compound as close as possible in the UMAP plot and the distance between the different compounds as far as possible. The area of the convex hull from Reviewer #3 is very enlightening, and a statistical

comparison of all samples is more convincing than highlighting some representative samples due to its comprehensiveness and objectivity. As described in response to Comment 2 of Reviewer #1, after careful consideration and inspired by Comment 3 of Reviewer #3, we think that **the distribution of relative Euclidean distances (RED)** is a better approach to show the difference between experimental and predicted spectra of compounds with different representations.

The RED values between experimental and predicted spectra have been calculated for all the 1,413 unique recall@1 molecules of FastEI using the UMAP dimensionality-reduced data of both the binning vectors and the embeddings. As shown in Fig. R7, all the RED of the binning vectors and the embeddings are visualized in their respective violin plots. It shows that the spectral embedding can narrow the RED between the predicted and experimental pair compared to the binning vector. The median of embedding RED values is 0.023, smaller than the median of binning RED values (0.026). The standard deviation of embedding RED is 0.069, which is only half of the standard deviation of binning RED (0.126). As stated by Reviewer #3, noisy bins weaken the power of discrimination. The Word2vec model removes noisy bins that can interfere the possible discrimination. It is clear that Fig. R7 is more convincing than the original Supplementary Fig. 7 (c, d). Hence, we replaced them with Fig. R7 in this revision to present the comparison results in a more comprehensive and objective manner.

Fig. R7^{††}. The violin plot of relative Euclidean distance (RED) between predicted and experimental spectral Word2vec embeddings and binning vectors of all 1413 molecules. M-RED represents the median of RED, and the SD-RED represents the standard deviation of RED.

Comment 4. P9 L30-49 “(a) NIST 2017, (b) ChEMBL 28”. In this study, 2 different training set are referred. One is f-NIST (232,826) as NEIMS training set, and another is f-NIST (232,826) plus f-ChEMBL (1,890,869) as Word2vec training set. This is not clearly mentioned in the main text, therefore still causing my misread. Authors require to mention this point in the method. Also, legend of Fig. 2b “Training set” is preferred to be more identical one, such as “NEIMS training set”.

Response: Thanks for pinpointing the misleading parts of the training sets. In fact, the NEIMS model was downloaded from its official repository without any further operation.

https://storage.googleapis.com/deep-molecular-massspec/massspec_weights/massspec_weights.zip.

The NEIMS training set, the NEIMS validation set, and the NEIMS test set were downloaded from the following URL:

https://github.com/brain-research/deep-molecular-massspec/tree/main/training_splits

^{††} This is Supplementary Figure 6c in the revised manuscript, and it is presented here for the convenience of reviewers.

A screenshot of this URL is shown in Fig. R8. Since we didn't train the NEIMS model, the "NEIMS training set" is only mentioned in Supplementary Fig. 1, and the "Datasets building for the libraries" subsection of the methods section in this study.

Fig. R8. A screenshot of the dataset split of NEIMS in its GitHub repository.

According to suggestions from Reviewer #3, to clarify the datasets, we have made the following two revisions in revised manuscript:

Firstly, we have adjusted Supplementary Figure 1. The revised Supplementary Figure 1 is provided here as Fig. R9 for the convenience of the reviewers. As shown in Fig. R9, we add the "NEIMS training set", "NEIMS validation set", and "NEIMS test set" to Fig. R9a. The datasets training set (232,826), validation set (11,496), and test set (11,499) are obtained by respectively filtering the datasets NEIMS_training set (237,108), NEIMS_validation set (11,499), and NEIMS_test set (11,600) with the five filtering rules. Then, we combined the training set, the validation set, and the test set to get the f -NIST for building the *in-silico* library. Fig. R9a can better show our actual data processing process.

Additionally, the predicted spectra of molecules in the training set (232,826) and f -ChEMBL (1,890,869) were merged into the **Word2vec training set** for Word2vec model training (Fig. R9c). We adjusted the description of the datasets in the method section. The datasets are described in two subsections: "Datasets for building the libraries" and "Datasets for FastEP" (Line 8-Line 108, Page 9).

Fig. R9.^{‡‡} Details of the datasets, libraries, and Word2vec model construction. **a** Detailed information on datasets used in FastEI. **b** Construction of the *in-silico* libraries. **c** Building of the Word2vec model.

As mentioned above, the training set was obtained by filtering the NEIMS training set, so the training set is taken as a subset of the NEIMS training set. When plotting Fig. R10, the ECFPs of the 232,826 molecules in the training set were used in practice. If the training set can cover the chemical space of the *f*-ChEMBL, the NEIMS training set would cover the space of the *f*-ChEMBL even better. As shown in Fig. R10, the training set covers the molecules randomly sampled from *f*-ChEMBL in the chemical space. Therefore, it is reasonable to apply the NEIMS model to predict the mass spectra of molecules in the *f*-ChEMBL, and the legend of Fig. R10 should be the "Training set" instead of the "NEIMS training set".

^{‡‡} This is Supplementary Figure 1 in the revised manuscript, and it is presented here for the convenience of reviewers.

Fig. R10^{§§} The visualization of the ECFPs of 240,000 molecules randomly selected from *f*-ChEMBL and 232,826 molecules from the training set by UMAP

Comment 5. P7 L19-20 recall@top x is 0%. I recommend to re-phrase it.

Response: Thanks to reviewer #3 for the recommendation on English expressions. We have re-phrased this expression “*the recall@top x is 0%*” to “*the correct result cannot be achieved*”.
(Line 19-Line 20, Page7)

^{§§} This is Fig. 2b in the revised manuscript, and it is presented here for the convenience of reviewers.

REVIEWERS' COMMENTS

Reviewer #1 (Remarks to the Author):

Thank you to the authors for their detailed follow-up and analysis. My questions/comments have been addressed.

A few small comments:

- Regarding Figure R2/Supplementary Figure 6 with the Relative Euclidean distances from the UMAP representation: I am not sure how to interpret differences in distribution of REDs as significant. Visual inspection of the UMAPs also does not suggest a big difference in distribution; but this could perhaps be improved with a different color selection. Perhaps the point could be made more strongly with a comparison to a random embedding/baseline distance.

To me, the contribution of the embedding representation is already made clearly by Figure R5/Supplementary figure 8, and makes the analysis in Figure R2 somewhat unnecessary.

- In the main manuscript: In my version, Table 1 has an a linebreak in Recall@10% making it somewhat difficult to read.

Reviewer #3 (Remarks to the Author):

The revised manuscript has addressed all of my major comments. Therefore, I support its publication from Nature Communications when the comments from reviewer#1 have been adequately addressed.

A typo is found in p8 L84 in the revised manuscript; close parenthesis expected like "...the discrimination). Therefore...".

Responses to reviewer comments

Reviewer #1:

Thank you to the authors for their detailed follow-up and analysis. My questions/comments have been addressed.

A few small comments:

Comment 1. Regarding Figure R2/Supplementary Figure 6 with the Relative Euclidean distances from the UMAP representation: I am not sure how to interpret differences in distribution of REDs as significant. Visual inspection of the UMAPs also does not suggest a big difference in distribution; but this could perhaps be improved with a different color selection. Perhaps the point could be made more strongly with a comparison to a random embedding/baseline distance. To me, the contribution of the embedding representation is already made clearly by Figure R5/Supplementary figure 8, and makes the analysis in Figure R2 somewhat unnecessary.

Response: Thanks for this professional comment from Reviewer #1. There are differences in the mean and standard deviation values of REDs from the UMAP plot in Figure R2/Supplementary Figure 6. However, the differences in the mean values were not so significant. Considering the contribution of the embedding representation is already made clearly in Figure R5/Supplementary Figure 8. Therefore, we have removed Supplementary Figure 6 and all related descriptions in the revised manuscript and Supplementary Information.

Comment 2. In the main manuscript: In my version, Table 1 has a linebreak in Recall@10% making it somewhat difficult to read.

Response: Thanks to reviewer #1 for pointing out this confusion. To make it clear, we deleted this linebreak in Recall@10% of Table 1.

Reviewer #3:

The revised manuscript has addressed all of my major comments. Therefore, I support its publication from Nature Communications when the comments from reviewer#1 have been adequately addressed.

A few small comments:

Comment 1. A typo is found in p8 L84 in the revised manuscript; close parenthesis expected like “...the discrimination). Therefore...” .

Response: Thanks to reviewer #3 for pointing out this typo. We have corrected it in the revised manuscript. (Line 69, Page5)